# Sick without signs. Subclinical infections reduce local movements, alter habitat selection, and cause demographic shifts
Marius Grabow [1,2] ✉, Wiebke Ullmann[3], Conny Landgraf[1], Rahel Sollmann[1], Carolin Scholz[1,3], Ran Nathan [4], Sivan Toledo [5], Renke Lühken [6], Joerns Fickel [1], Florian Jeltsch [3], Niels Blaum [3], Viktoriia Radchuk [1], Ralph Tiedemann [7] & Stephanie Kramer-Schadt [1,2]

In wildlife populations, parasites often go unnoticed, as infected animals appear asymptomatic. However, these infections can subtly alter behaviour. Field evidence of how these subclinical infections induce changes in movement behaviour is scarce in free-ranging animals, yet it may be crucial for zoonotic disease surveillance. We used an ultra-high-resolution tracking system (ATLAS) to monitor the movements of 60 free-ranging swallows every 8 seconds across four breeding seasons, resulting in over 1 million localizations. About 40% of these swallows were naturally infected with haemosporidian parasites. Here, we show that infected individuals had reduced foraging ranges, foraged in lower quality habitats, and faced a lowered survival probability, with an average reduction of 7.4%, albeit with some variation between species and years. This study highlights the impact of subclinical infections on movement behaviour and survival, emphasizing the importance of considering infection status in movement ecology. Our findings provide insights into individual variations in behaviour and previously unobservable local parasite transmission dynamics.

Parasitic infections are a growing concern for species' and ecosystem health and are an emerging cause of biodiversity loss[1,2]. These infections often go undetected because infected animals do not show clear signs of being sick, masking the true extent of their impact[3]. These subclinical infections – where pathogens are present without causing apparent symptoms - can significantly affect an animal's fitness and may persist long before symptoms become noticeable. By inducing physiological costs for immune defences, chronic parasite infections continuously drain the energetic reserves of their hosts[4], which are otherwise allocated to activities like foraging or mating[5]. This can alter host behaviour[6,7] and eventually reduce their ability to survive and reproduce, impacting their Darwinian fitness[8].

Movement, as a plastic behavioural trait, directly links animals' movement decisions to energy use[9,10] and is well known to be impacted by severe infections[11,12]. In the presence of parasites, energetic balances shift as individuals have to allocate resources towards costly immune functions[4]. This can lead to altered movement behaviour, e.g. diminished migratory ranges e.g.[13] or decreased daytime activity in foraging e.g.[14]. Such alterations can, in turn, influence various other plastic traits, including morphological

ones[11]. However, most studies in movement ecology lack access to health data and implicitly assume individuals are performing at their optimal capacities[3]. Yet, this oversight limits our understanding of parasites as key drivers of individual movement variation and their impacts on fitness[15]. Moreover, failing to include parasitic infections in movement studies impedes accurate predictions of disease outbreaks[16,17]: Parasite-induced behavioural shifts, like sickness behaviour, influence transmission dynamics by affecting contact rates and hence the spread of diseases within animal communities[18,19].

In spite of the ecological importance of host-parasite dynamics, incorporating parasitic infections into movement ecology studies proves challenging – especially for asymptomatic infections, i.e. those without any visible signs of declining health. Biologging has been used to unravel behaviour, cognition and social interactions of animals[20–23], but not yet to uncover subtle impairments of animal movements by non-visible parasite infections. These subtle impairments are likely to affect daily activity budgets and fine-scale movements, spatial memory, navigation and orientation capabilities, and direct and indirect communication.

[1]Leibniz Institute for Zoo and Wildlife Research, Alfred-Kowalke-Straße 17, Berlin, Germany. [2]Institute of Ecology, Technische Universität Berlin, Rothenburgstr. 12, Berlin, Germany. [3]Plant Ecology and Nature Conservation, Universität Potsdam, Zeppelinstraße 48A, Potsdam, Germany. [4]Movement Ecology Laboratory, Department of Ecology, Evolution & Behavior, The Hebrew University of Jerusalem, Jerusalem, Israel. [5]Blavatnik School of Computer Sciences, Tel Aviv University, Tel Aviv, Israel. [6]Bernhard Nocht Institute for Tropical Medicine, Bernhard-Nocht-Straße 74, Hamburg, Germany. [7]Evolutionary Biology/Systematic Zoology, Institute of Biochemistry and Biology, University of Potsdam, Karl-Liebknecht-Straße 24-25, Potsdam, Germany. ✉e-mail: grabow@izw-berlin.de

While the lack of wildlife health data already poses a substantial obstacle, the primary challenge in detecting the impacts of infection arises from technical limitations in animal tracking that traditionally restricted studies to larger spatiotemporal scales[20]. These large spatiotemporal scales prevented the detection of immediate parasite impacts on host behaviour that often occur within hours or days after infection e.g.[24]. Recent technological advances in high-resolution tracking, such as the lightweight ATLAS (Advanced Tracking and Localisation of Animals in Real Life Systems) tracking technology[21], allow us to overcome these limitations and detect even nuanced differences in behaviour that are invisible at larger scales[20]. This renders the study as one of the first uncovering direct behavioural implications of subclinical infections.

Here we seek to unravel the impacts of asymptomatic infections on plastic phenotypic traits, like movement behaviour, and morphological traits, like body condition, and to investigate how these changes scale up to demographic rates, including survival. As a proof of concept, we study philopatric wild swallow species naturally infected with avian blood parasites. The populations of colony-breeding barn swallows (*Hirundo rustica*) and house martins (*Delichon urbicum*) – henceforth collectively referred to as 'swallows'– constitute a seasonally closed system facilitating demographic studies and long-term monitoring of this host-parasite system[25]. The long-distance migratory behaviour of swallows between breeding and wintering grounds may increase their exposure to parasites[26–28], including common avian blood parasites (haemosporidians), which cause avian malaria and are transmitted by blood-sucking vectors like mosquitoes[29]. Avian blood parasites are widely prevalent[30] and have well-documented effects on host physiology and energy reserves due to their destruction of red blood cells, potentially leading to anaemia and impairing oxygen transport to tissues and organs[29]. Evidence suggests that these blood parasites are likely to persist through the breeding season, posing a continuous immune challenge on top of the already energetically costly demands of breeding[29]. Swallows, being obligatory aerial insectivores, must hunt ephemeral and patchily distributed prey, which is reflected in their distinctive foraging behaviour[31]. Given their high metabolic rates relative to body size[32], swallows are likely to exhibit responses to parasitic infection in their behaviour and morphological traits, e.g. reduced body weights due to the energetically costly immune challenges[4].

We predicted that infected individuals show decreased mobility due to increased resting behaviour, translating to reduced foraging ranges. Consequently, we expected these changes to alter habitat selection patterns, with infected birds being unable to access energy-rich habitats and thus showing no preferences for specific habitat types, but rather using whatever is available. We anticipated these changes to be reflected in altered morphological traits, specifically decreased body condition as a measure of physical health. We further predicted that such alterations would have negative effects on survival rates when compared to non-infected conspecifics. To study parasite-induced behavioural shifts, we utilized the established high-resolution tracking system ATLAS[21] that collated >1 million localizations of 60 tagged and blood-sampled swallows at second intervals. We used advanced movement models to derive spatial and behavioural patterns and paired these with survival data obtained through capture-mark-recapture hierarchical Bayesian population models of 663 blood-sampled swallows across four breeding seasons (Fig. 1).

## Results
### Animal capturing and parasite prevalence
Across four breeding season years, we captured and measured 389 individual barn swallows and 274 individual house martins, with females constituting 51% of the barn swallows and 58% of the house martin captures. On average, individuals were captured 1.46 ± 0.74 (mean ± SD) times per day and 2.11 ± 1.28 (mean ± SD) times in the same year, i.e. 14 days later. Individuals that were tagged and recaptured after 14 days were not tagged again. Blood parasite prevalence varied between years but the effect was not significant in barn swallows ($\chi^2(6) = 8$, $p = 0.238$) and house martins

($\chi^2(9) = 12$, $p = 0.213$), with an average of 37.9% ± 24.4% (mean ± SD) of individuals infected with at least one of the three tested parasite genera (Table 1).

### Parasite-related effects on foraging ranges
We recorded 1,318,683 locations of 97 tagged individuals that were reduced to 76 individuals with 993,977 locations after filtering for outliers. On average, individuals were tagged for 71.76 hours ± 77.60 (mean ± SD). Swallows foraged close to their colonies (Supplementary movie S1), with mean maximum displacement distances of 2484 m ± 1269 (mean ± SD). Mean movement speeds were significantly higher in non-infected individuals (18.00 km h$^{-1}$ ± 9.12), compared to infected individuals (12.57 km h$^1$ ± 5.72), (Welch t-test: t(27.743) = −2.777, $p < 0.001$). Throughout the study, all individuals exhibited stationary behaviour, best described by Ornstein-Uhlenbeck motion models that incorporated foraging behaviours (OUf, Supplementary material S6). Additionally, our analysis confirmed that the estimated foraging ranges (measured as 95% autocorrelated Kernel Density Estimate; aKDE) were not influenced by the duration of tracking, indicating that the tracking duration was sufficient for describing foraging ranges (Supplementary material S6). House martins showed a tendency towards larger foraging ranges compared to barn swallows ($X^2$-Inverse Gaussian hierarchical model, $p = 0.059$) (Table 2). On average, foraging ranges (95% aKDE) were 1.61 [CI95% 0.48–4.27] times larger in non-infected barn swallows, and 1.44 [CI95% 0.62–2.74] times larger in house martins; relative effect sizes calculated following Fleming et al.[33] (Fig. 2a). Specifically, non-infected barn swallows had an average foraging range of 2.50 km² [CI95% 1.282–4.383], while infected individuals used 1.31 km² [CI95% 0.583-2.553]. For house martins, non-infected individuals used 5.13 km² [CI95% 2.714–8.778], whereas infected individuals used 3.87 km² [CI95% 2.487–5.761]. This trend was consistent across the smaller 50% aKDE core ranges (Table 2). Non-infected individuals of both species consistently showed higher coefficients of variation (Table 2), i.e. variation of foraging range sizes of infected individuals was decreased. On average, non-infected barn swallows showed a range overlap of 43.5% [95% CI: 0.36–0.50], while infected conspecifics had an overlap of 90.7% [95% CI: 0.87–0.94]. The ranges of non-infected house martins overlapped by 48.1% [CI95%: 0.36–0.53], and those of infected individuals by 56.6% [CI95%: 0.54-0.59] (Supplementary material S6).

### Parasite-related effects on foraging and resting behaviour
The Hidden Markov Models (HMM) discerned notable differences in the behavioural states of infected versus non-infected individuals (Fig. 2b). Infected individuals of both species foraged significantly less; foraging was significantly reduced by 15% in barn swallows (Welch t-test: t(2.01) = 10.10, $p = 0.010$) and significantly reduced by 8% in house martins (Welch t-test: t(118.66) = 56.30, $p < 0.001$). Likewise, resting was significantly increased in infected individuals of both species; by 17% in barn swallows (Welch t-test: t(2.01) = −11.45, $p = 0.007$), and by 10% in house martins (Welch t-test: t(120.96) = −70.00, $p < 0.001$). Commuting behaviour, however, underwent only minimal changes in both species (Fig. 2b). Similarly, state switching probabilities towards resting were higher in infected individuals (Supplementary material S7). We were unable to classify foraging behaviour using animal tracking data of coarser resolution 30-minute intervals as often used in tracking studies (Supplementary material S8).

### Parasite-related effects on habitat selection
The integrated Step Selection Functions (iSSF) identified significant differences in habitat use during foraging behaviour between infected and non-infected individuals of both species (Fig. 2c). In general, non-infected individuals of both species avoided foraging in semi-natural environments (Table 3), while infected ones did not show preferences for specific habitat types (Fig. 2c). Notably, non-infected barn swallows significantly avoided foraging over agricultural habitats whereas infected barn swallows exhibited no distinct habitat preference or avoidance of agricultural habitats relative to human-related habitats (Table 3). Likewise, house martins significantly

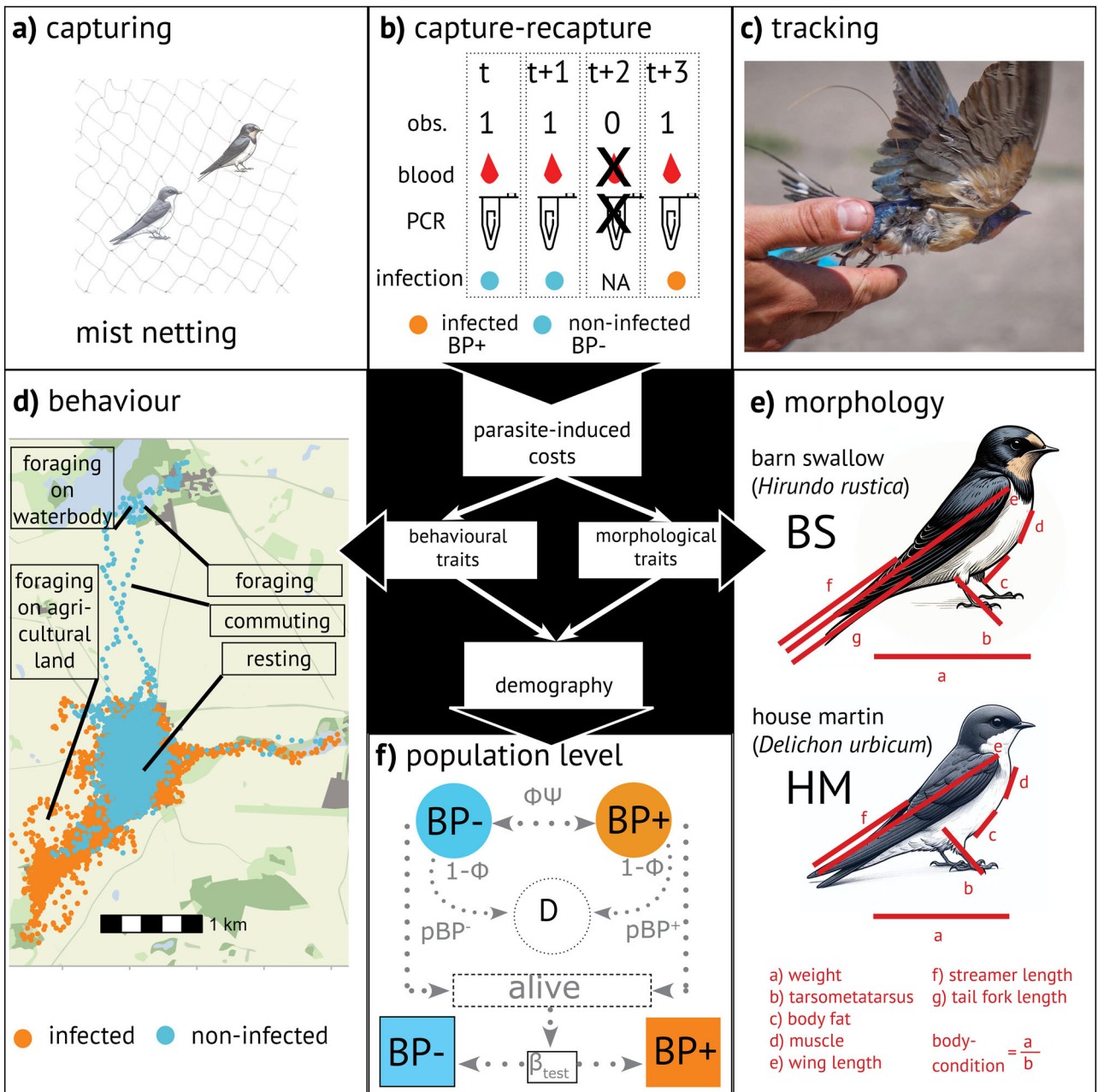

**Fig. 1 | Overview of hypothesis, data collection, and analyses. Centre panel:** Main hypothesis: Parasites alter resource allocation in their hosts, leading to changes in such plastic traits as morphological traits and movement behaviour, with potential consequences for demography. **a** Capturing method for birds: Mist netting using a structured sampling design with equal sampling effort. **b** Scheme of Capture-mark-recapture (CMR) design, including multiple observations (obs.) across years t, when blood samples were collected and analysed via polymerase chain reaction for blood parasite infection. In case of non-recaptures (see example at t + 2) no blood samples would be collected. **c** Barn swallow (BS) tagged during CMR with lightweight ATLAS tag (0.125 Hz) for movement analyses. **d** Exemplary movement tracks (Supplementary movie S1), analyses of movement behavioural states (foraging, commuting, and resting via Hidden-Markov model; HMM), and habitat selection examples during foraging (via integrated Step-Selection function, iSSF). **e** Morphological traits of barn swallows (BS) and house martins (HM) measured during the CMR. **f** Schematic representation of multi-event model for analysing survival by accounting for uncertainty in blood parasite (BP) infection status; BP+, BP-, and D (round circles) describe the true state of each individual, namely infected with BP, non-infected, and dead, respectively. Transitions between BP+ and BP- are explained as survival (Φ) and state transition (ψ), or death (1- Φ). Observations (pBP + and pBP-) differ between infected and non-infected individuals, dead individuals are never captured. Infection status of subclinical infection can only be revealed by polymerase chain reaction, i.e. the testing probability (β_Test).

avoided agricultural habitats while infected house martins showed no habitat preference relative to human-related habitats (Table 3).

**Morphological traits**

We found no evidence for difference in body condition (measured as weight divided by tarsometatarsus; Fig. 1e) between infected and non-infected individuals (barn swallows: $F_{(1, 329)} = 1.533$, $p = 0.217$; HM: $F_{(1, 346)} = 0.788$, $p = 0.375$). This was also confirmed by analysing all morphological traits (Fig. 1e) via Principle Component Analysis (Supplementary material S10). Furthermore, we found no evidence that morphological traits would be altered over the timescale of active parasitic infection in individuals, as both groups showed only minor alterations in morphological traits over the period of the active parasitic infection (Supplementary material S10). However, morphological traits

**Table 1 | Description of the collected data, prevalence of avian haemosporidian parasites in the studied barn swallow (BS) and house martin (HM) populations in four study years prior to filtering movement data**

| Species | Year | Captured (n) | Blood sampled (n) | Tracked (n) | Plasmodium ssp. prevalence | Haemoproteus ssp. prevalence | Leucocytozoon ssp. prevalence | Overall infection prevalence |
|---|---|---|---|---|---|---|---|---|
| BS | 2020 | 141 | 6 | 1 | 33.33% | 0% | 0% | 33.33% |
| | 2021 | 135 | 13 | 0 | 15.38% | 30.77% | 15.38% | 75.56% |
| | 2022 | 165 | 90 | 11 | 24.44% | 2.22% | 8.89% | 30.00% * |
| | 2023 | 144 | 140 | 36 | 10.34% | 0% | 1.38% | 11.72% |
| | **sum** | **585** | **249** | **48** | - | - | - | - |
| HM | 2020 | 133 | 7 | 0 | 14.29% | 28.57% | 14.29% | 57.14% |
| | 2021 | 125 | 37 | 1 | 27.03% | 51.35% | 2.70% | 75.68% * |
| | 2022 | 113 | 69 | 12 | 24.63% | 36.23% | 8.70% | 59.42% * |
| | 2023 | 129 | 106 | 36 | 4.72% | 8.49% | 0% | 11.32% |
| | **sum** | **500** | **219** | **49** | - | - | - | - |

The overall infection prevalence indicates the percentage of the population infected with any parasite, noting that some individuals had co-infections with the same or different parasite genera, which may cause overall prevalence to differ from the sum of individual genus prevalences (as indicated by *).
Bold values highlight significant effects.

**Table 2 | Estimated foraging ranges (as autocorrelated Kernel Density Estimate; aKDE) for barn swallows (BS) and house martins (HM)**

| Population | aKDE | Mean estimate [95% CI] | Coefficient of variation (CoV) [95% CI] | Relative effect size [95% CI] |
|---|---|---|---|---|
| BS$_{non-infected}$ | 95% | 2.50 km² [1.282–4.383] | 1.75 [1.06–2.45] | 1.61 [0.48–4.27] |
| BS$_{infected}$ | | 1.31 km² [0.583–2.553] | 0.67 [0.19–2.44] | |
| BS$_{non-infected}$ | 50% | 0.40 km² [0.174–0.768] | 2.11 [1.17–3.05] | 2.18 [0.61–5.76] |
| BS$_{infected}$ | | 0.18 km² [0.070–0.390] | 0.79 [0.21–1.41] | |
| HM$_{non-infected}$ | 95% | 5.13 km² [2.714–8.778] | 2.00 [1.00–3.00] | 1.44 [0.62–2.74] |
| HM$_{infected}$ | | 3.87 km² [2.487–5.761] | 1.33 [0.71–1.96] | |
| HM$_{non-infected}$ | 50% | 0.89 km² [0.326–1.920] | 2.00 [1.00–3.01] | 1.67 [0.53–3.97] |
| HM$_{infected}$ | | 0.43 km² [0.209–0.779] | 1.41 [0.337–3.249] | |

The first column indicates the two populations that were compared via χ2-Inverse Gaussian hierarchical models. Different aKDE sizes (50% and 95%) represent core and foraging ranges of the population with the mean estimate stated for each population [95% Confidence intervals of the given aKDE]. Relative effect sizes are following Fleming et al.[33]. Estimates represent mean values, and square brackets refer to 95% confidence intervals.

played an important role in explaining survival probabilities for infected barn swallows (see 2.5).

**Parasite-related effects on demography**

Model selection yielded different models for each species, both were distinct to the next best model (barn swallows: ΔWAIC: 4.599; HM: ΔWAIC: 8.773). We report all Bayesian estimates as mean and 89% Credible Intervals following Kruschke[34], who advocated for this approach in Bayesian statistics to avoid overconfidence and balance uncertainty with precision in interval levels.

The detection probability (p), i.e. the probability that an individual gets captured or re-captured via mist-nets, of infected individuals of both species was substantially lower compared to their non-infected conspecifics (Fig. 3a; model selection Supplementary material S11). In barn swallows, the mean detection probability was 0.343 [89% CI: 0.295-0.389] for non-infected and 0.212 [89% CI: 0.140-0.315] for infected individuals; in house martins, the mean detection probability was 0.638 [89% CI: 0.577-0.701] for non-infected and 0.537 [89% CI: 0.461-0.613] for infected individuals. The difference in survival probabilities between infected and non-infected individuals (Fig. 3b) was larger for house martins than barn swallows. Specifically, the average survival probability for non-infected barn swallows was 0.481 [89% CI: 0.383-0.585], whereas it was 0.433 [89% CI: 0.260-0.622] for the infected ones. In the case of house martins, non-infected individuals had a mean survival probability of 0.455 [89% CI: 0.235-0.725], while the infected ones had a mean of 0.356 [89% CI: 0.157-0.547]. For barn swallows, the proportion of posterior samples in which the difference in survival between

infected and non-infected individuals was below 0 (i.e. evidence in favour of lower survival probabilities of infected individuals), was 62.8%; in house martins, this evidence was 76.8%, 72.0%, and 38.8% in 2021, 2022, and 2023, respectively (Supplementary material S13). For barn swallows, individuals with larger body condition had a higher survival probability, however, model predictions towards extremely large or small body conditions were based on few observations only (Fig. 3c). The house martin model revealed annual fluctuations in survival probabilities (Fig. 3d; Supplementary material S12). We estimated relatively high probabilities of clearing the infection from the bloodstream, as determined by polymerase chain reaction (Supplementary material S12). It is important to note that this analysis only addressed the clearance of the disease from the bloodstream and did not provide information on potential chronic disease states. The simulation study revealed low error and biases in parameter estimates; 92% of all estimates were within the 89% CI, and estimated mean survival parameters deviated on average 4.32% ± 3.88 (mean ± SD) from the true value (Supplementary material S14).

**Discussion**

In this study, we tracked free-ranging individuals of two sympatric swallow species at the high temporal resolution, revealing notable behavioural differences between non-visibly infected and non-infected individuals—differences that would remain undetected with coarser resolution data. By combining these behavioural shifts with demographic studies, we found evidence that the altered movement behaviour of infected individuals is accompanied by effects at the population level, with blood parasites

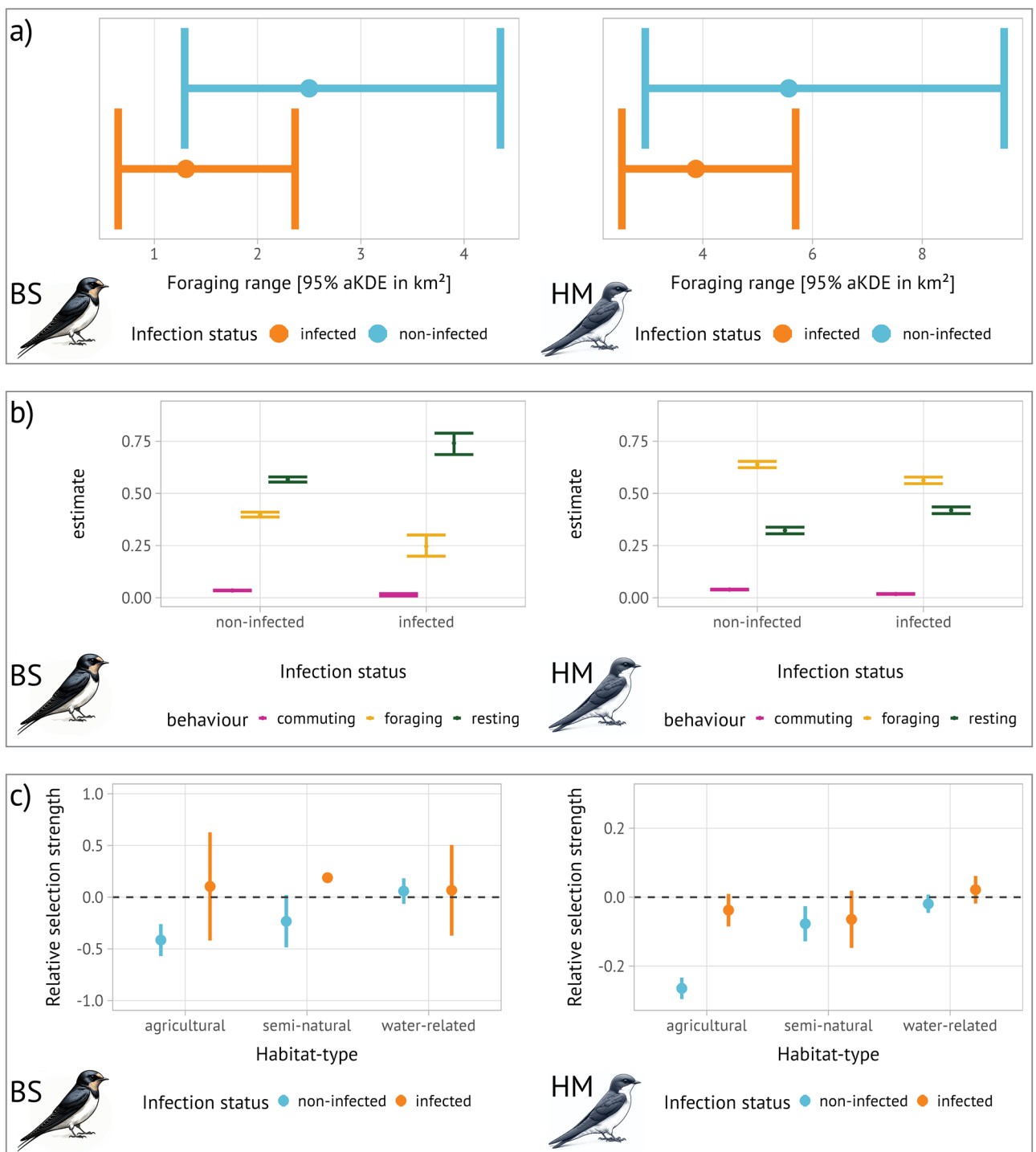

**Fig. 2 | Parasite-related changes in movement behaviour. a** Foraging ranges of barn swallows (BS; left panel) and house martins (HM; right panel) measured as autocorrelated Kernel Density estimates; aKDE. **b** Behavioural classification of movement indicates more time allocated to resting behaviour at the cost of reduced foraging behaviour in infected individuals compared to non-infected ones. **c** Integrated Step Selection Function (iSSF) analysis during 'foraging' relative to foraging in human settlements (the baseline habitat type). Negative relative selection strength (RSS) indicates significant avoidance of agricultural land during foraging by non-infected individuals, whereas infected individuals forage over these habitats. Similarly, non-infected individuals show increased habitat preference for foraging over habitat associated with water bodies. Throughout panels **a–c**, points represent mean values, and lines refer to 95% confidence intervals.

reducing survival. These subclinical effects of parasites may also act in synergy with other stressors, such as climate change[35,36], quietly affecting animal populations. Our key finding that infected individuals foraged less and rested more was consistent across two distinct methodologies: behavioural movement analysis using high-resolution tracking data and demographic modelling using capture-recapture and accounting for detection rates that were lower in infected individuals. Such consistency across approaches suggests that observed behavioural changes are due to infection and that they can indeed be measured at a fine spatiotemporal scale. Moreover, it implies that such behavioural changes contribute to the

**Table 3 | Summary of fixed and random effects from generalized linear mixed-effects models (GLMMs) for barn swallows (BS) and house martin (HM), analysing the relationship between habitat use and infection status**

| Species | Effects | Predictor variable | Infection status | Coefficient β | Se | P | Std.dev |
|---|---|---|---|---|---|---|---|
| BS | fixed | agricultural | non-infected | −0.400 | 0.077 | **<0.001** | |
| | | | infected | 0.124 | 0.263 | 0.638 | |
| | | semi-natural | non-infected | −0.214 | 0.129 | 0.097 | |
| | | | infected | −0.166 | 0.460 | 0.718 | |
| | | water-related | non-infected | 0.073 | 0.058 | 0.209 | |
| | | | infected | 0.069 | 0.210 | 0.746 | |
| | random | Intercept: tag_id | | | | | 0.082 |
| | | Intercept: step_id | | | | | 0.402 |
| | | Slope: agricultural \|tag_id | | | | | 0.269 |
| | | Slope: semi-natural \|tag_id | | | | | 0.453 |
| | | Slope: water-related \| tag_id | | | | | 0.215 |
| HM | fixed | agricultural | non-infected | −0.266 | 0.016 | **<0.001** | |
| | | | infected | −0.034 | 0.024 | 0.162 | |
| | | semi-natural | non-infected | −0.082 | 0.026 | **0.002** | |
| | | | infected | −0.034 | 0.043 | 0.076 | |
| | | water-related | non-infected | −0.021 | 0.013 | 0.122 | |
| | | | infected | 0.024 | 0.020 | 0.076 | |
| | random | Intercept: tag_id | | | | | <0.001 |
| | | Intercept: step_id | | | | | <0.001 |

Models used 'true' steps as the response variable, in relation to random steps, utilizing a Poisson distribution. We report the estimated coefficients (β), standard errors (SE), and associated $p$-values ($p$) for fixed effects that represent foraging on different habitat types. Random effects describe the variation between individuals (tag_id) and individual steps (step_id), with corresponding standard deviation (SD). Random slopes for each habitat type (in BS) illustrate inter-individual variability in habitat selection. Random slope models for House martins (HM) did not coverge.
Bold values highlight significant effects.

reduced capture rates observed with the CMR study. As opposed to our analysis, previous studies on the movement of parasitized hosts often focused on large spatiotemporal scales, such as long-distance dispersal or migration e.g.[26,37,38]. Though a meta-analysis on movement patterns at coarse migratory scales also reported a reduction in the movement of infected individuals[39].

Our approach introduces the ability to discern even subtle differences in local movements, thereby enabling the analysis of behaviours, including foraging patterns, which remain undetectable at coarser resolutions. Although the immense potential of high-resolution tracking has been demonstrated previously[20], for instance in studies of cognitive behaviour[21,40], integrating infection status into high-resolution movement analysis has not yet been explored and offers a new pathway for incorporating movement behaviour into disease ecology and epidemiology.

We revealed that a lower detection probability of infected individuals can be linked to decreased mobility, consistent with prior expectations regarding capture rates in wild populations[27]. Indeed, a reduction of movement due to infection is an intuitive and a well-documented phenomenon observed in various species[41,42], including humans[43]. Previous studies have associated reduced capture rates with decreased movements due to high parasitaemia levels of avian blood parasites, which lead to anaemia[44]. Our findings extend this by evidencing that parasite-induced resting behaviour subsequently affects an individual's foraging range, restricting it to smaller areas around breeding colonies and lower-quality foraging habitats. Regardless of the underlying mechanism, whether adaptive sickness behaviour[42] or a physiological constraint due to infection[29], our findings highlight ecological implications. Specifically, the reduced-ranging behaviour in infected individuals is indicative of a change in habitat selection patterns.

It is crucial to recognize that small activity ranges are not necessarily disadvantageous for individuals. From the perspective of optimal foraging theory[45], individuals with smaller foraging ranges might be optimizing their energy use by foraging closer to their colony, thereby minimizing the energetic expenditure associated with flight. However, this strategy relies on the availability of suitable foraging habitats around the colony. Should the surrounding habitats be less favourable, animals need to travel further to fulfil their energetic needs or cope with suboptimal foraging[46,47]. If infection hinders individuals from reaching distant but profitable habitat patches, a reduction in daily foraging ranges may indicate substantial impairment.

Despite observing a clear tendency towards decreased foraging ranges in infected individuals of both species, results were quite variable among individuals, with some infected individuals even exhibited larger foraging ranges than their non-infected conspecifics. This could indicate intraspecific differences in response to parasitic infection explained by parasite loads[48], host behaviour–parasite feedbacks[49], or genetic variance[50]. Interestingly, non-infected individuals showed greater variation in foraging ranges, suggesting that infected individuals indeed faced parasite-induced consequences restricting their foraging behaviour.

We revealed that non-infected individuals significantly avoided foraging over agricultural areas, where lower insect abundances are typically anticipated[51]. Contrarily, infected ones neither showed strong preferences for not avoided any specific habitat types, including agricultural habitats. Given that aerial insects are often patchily distributed, this strongly suggests infected individuals would then forage in apparently less energy-rich habitats. Previous research has linked foraging of aerial insectivores directly to reproductive performance, indicating that foraging in deteriorated habitats has direct implications on fledging success[52]. With current trends in land use deterioration and declining insect populations[53], these effects may become more pronounced. Such changes could force animals, when able, to travel longer distances to increasingly scarce suitable habitats. Conversely, the presence of high-quality habitats near breeding locations could mitigate the adverse effects of infection, potentially decreasing host health impairment. This underscores the importance of habitat management practices that ensure sufficient food resources, such as high aerial insect abundances and diversity[54]. The finding that infected individuals more frequently utilize agricultural lands is particularly interesting from a One Health perspective.

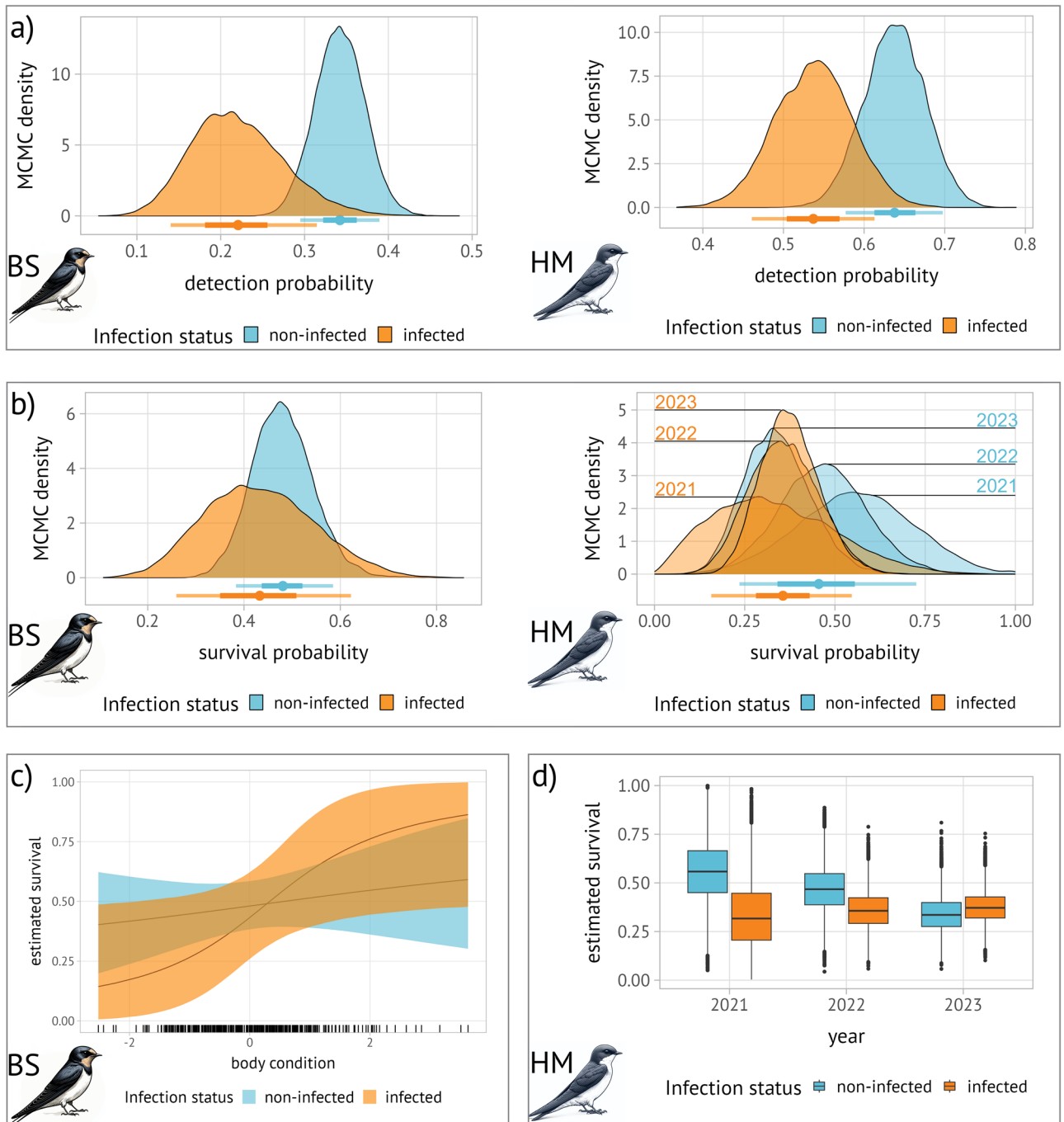

**Fig. 3 | Parasite-related demography of two studied host species.** Posterior distributions of non-infected and infected barn swallows (BS, left panel) and house martin (HM, right panel); **a** Posterior distributions for detection probability. **b** Posterior distributions for survival probability. In panels **a** and **b**, points represent mean values, and lines refer to 50% (thick line) and 89% credible intervals (thin line). **c** Predicted effect of body condition on survival in barn swallows by infection status, according to the selected model. In **c**, thin black lines represent parameter estimates, and shaded areas refer to 89% credible intervals. **d** Predicted effect of year on survival in HM by infection status, according to the selected model. In panel d, boxes represent the IQR, lines inside the box the median, and whiskers the 1.5 IQR from the first and third quartiles, respectively.

Although avian haemosporidian infections are not pathogenic to livestock, there is growing evidence that spill-over events are becoming more common in human-altered ecosystems[2], increasing the risk of zoonotic diseases. Notable examples include recent avian influenza outbreaks affecting dairy cows[55] and the zoonotic transmission of Hendra virus from bats to livestock[56], both eventually affecting humans.

While our findings provide valuable insights into the relationship between parasitic infections and movement patterns, it is important to consider the potential causal mechanisms underlying these associations. In our correlational study, there are three potential pathways for how these associations could emerge: (1) infection alters movement patterns, (2) underlying fitness deficiencies cause both reduced survival and movement, and (3) individual movement behaviours predict exposure to infection. We argue that the first causal relationship—infection altering movement patterns—is strongly supported due to the nature of the host-parasite interaction. Avian blood parasites hinder oxygen transport to

tissues, resulting in anaemia and physiological limitations[29,57], as documented in previous experimental studies[44,58]. There is limited support for the second hypothesis, which posits that underlying fitness deficiencies predict both reduced survival and movement, as we found no significant differences in morphological traits between infected and non-infected individuals. The third hypothesis, however, that movement behaviours predict exposure to infection, warrants further investigation. One argument against this hypothesis is that individuals typically rest at their colonies, where a dilution effect could potentially reduce individual infection risk, although current evidence of the dilution hypothesis in avian haemosporidians remains inconclusive[59,60]. On the other hand, swallows, as aerial insectivores, may directly prey on potential disease vectors[61–63], potentially lowering the infection risks close to their colonies. Further evidence to discriminate between the first and the third hypothesis could have been collected by comparing the movements of individuals that changed their infection status and were re-tagged. Unfortunately, we did not observe such cases, further highlighting the need for experimental research to test these hypotheses more rigorously[27]. For example, randomly assigning individuals into different treatments groups, performing infection challenges, and assessing movement behaviour. Such experiments could conclusively determine whether movement behaviour is impacted by infection, though they come with ethical considerations and feasibility challenges.

We found no direct evidence linking parasite infection to changes in morphological traits. This outcome was unexpected, as we had anticipated that the exploitation of host resources by parasites would manifest not only in movement behaviour, as observed, but also in body measures. On the other hand, studies that involved individuals experimentally infected with blood parasites have demonstrated that survivors did not significantly change body mass post-infection, despite considerable alterations in their behaviour[58]. One possible explanation is that sickness behaviour[42], characterized by increased resting, helps individuals conserve energy reserves, such as fat, thereby maintaining their body condition. This hypothesis aligns with our observations of infected individuals resting more. However, infected individuals had a higher mortality probability, suggesting that they died to causes other than direct starvation effects, such as long-term effects following infection[64,65], increased predation risk e.g.[44], or during migration e.g.[13].

In contrast, our demographic model revealed that barn swallows with better body condition had higher survival probabilities, regardless of infection status, implying that these individuals could partly mitigate impacts of parasites. This finding corroborates previous studies on correlative effects between body condition and survival[66,67], although we cannot make the direct link to movement behaviour in this case. Despite using lightweight tracking devices, our movement study is biased towards tagging only the heaviest individuals of the population, as per animal welfare permits. Thus, generalizing our findings of the tracking study to the entire population requires caution as we could not differentiate patterns in movement behaviour associated with body condition. Building on this previous knowledge, we presume that effects might be even more pronounced in smaller or lighter animals with poorer body condition, given their higher energetic demands, and thus increased activity, relative to their body sizes[68].

Decreased survival is commonly expected in many host-parasite systems. Yet, in subclinical blood parasite infections this effect is often less evident, particularly when host and parasite have a long co-evolutionary history[64,69,70]. Our study indicated that blood parasite infection correlated with lower year-to-year survival. Although differences of 5 – 10% in annual survival probability may seem subtle, they are biologically meaningful and can significantly impact population dynamics over time[71]. Notably, parasites may not directly cause death[72]; rather, reduced survival could stem – at least partly – from the diminished foraging activity found in our analyses of the movement data. Furthermore, parasite-induced resting could heighten predation risks of parasitized individuals, as observed in an experimental study by Mukhin et al.[44]. Here, we focused on analysing year-to-year

survival, and therefore could not pinpoint death locations, making us unable to conclusively assess causes of death. Likewise, our analyses of survival rates require caution although our simulation study indicated that our models were capable of obtaining relatively unbiased estimates given the sample sizes. The evidence in favour of lower survival of infected individuals varied between years, with one year even yielding higher survival probabilities of infected individuals (house martins in 2023). Nevertheless, increased survival may still compromise fitness if infected individuals forego reproduction to enhance immediate survival, potentially diminishing overall fitness. However, since we lack data on reproduction, this hypothesis requires further investigation.

Previous research in captive songbirds suggests that parasitaemia, or parasite load, more accurately reflects physiological impacts than a dichotomous infection status[73]. However, assessing parasitaemia in a free-ranging population is challenging due to the dynamic nature of avian blood parasites that would require multiple blood samples within a few days[29]. In our study, we focused on detecting parasites in the bloodstream, acknowledging the risk of missing chronic disease states where parasites are tissue-bound and may lead to relapses[74,75]. Consequently, individuals experiencing post-sampling relapses may be misclassified regarding their infection status. This potential misclassification could reduce the observed survival differences between groups, leading us to underestimate the impact of parasites on survival, making our estimates rather conservative.

While our study provides valuable insights into behavioural states and step selection, it is important to acknowledge the limitations of our modelling approach. Specifically, we performed the analyses in two steps, by first fitting HMMs and then applying a Welch test to assess whether the infected and non-infected individuals differ in their behavioural states. Such a two-step procedure cannot fully propagate the uncertainty associated with the parameter estimates obtained from the HMM. Ideally, a hierarchical model should be used to assess behavioural states at the individual level and directly compare them between the infection groups. However, current methods for incorporating random effects into HMMs are computationally demanding and carry the risk of overfitting[76,77]. Similarly, recent advancements that integrate HMMs with step selection functions (SSFs) into a combined framework can account for uncertainty in behavioural states and state-dependent habitat selection[78], but these approaches also come with significant computational costs. This highlights the need for continued advancements in developing statistical modelling frameworks to analyse animal movement data.

In conclusion, our study demonstrates that impacts of subclinical infections can lead to reduced daily foraging ranges, altered behaviours, and changes in habitat selection during foraging. Only high-resolution data could reveal these effects. Given the critical role of movement in energy balance, we suggest that these parasite-induced behavioural changes likely impair resource acquisition, contributing to the observed decrease in survival probabilities at the population level. Overall, our study offers a refined perspective on incorporating parasites in movement ecology, highlighting both potential sources of individual variation and their links to vital ecological processes that influence survival.

## Methods
### Study area
We conducted our study in northeast Germany (N 53.38°, E 13.75°) close to the city of Prenzlau (Fig. 4a-b), a predominantly agricultural area (68% of land) with scarce natural structures like forests and waterbodies, constituting about 5% and 6% of the landscape, respectively[79]. The landscape features patchily distributed water-resources within the agricultural matrix, serving as foraging patches for insectivorous species e.g.[22]; one example are the ephemeral waterbodies ("kettle holes"). We collated habitat types based on the colour-infrared biotope types via aerial scans[79] into four distinct habitat types (Fig. 4b): human-related habitat (built-up areas), agricultural, semi-natural, and water-related (kettle holes, swamps and adjacent grasslands; Supplementary material S2).

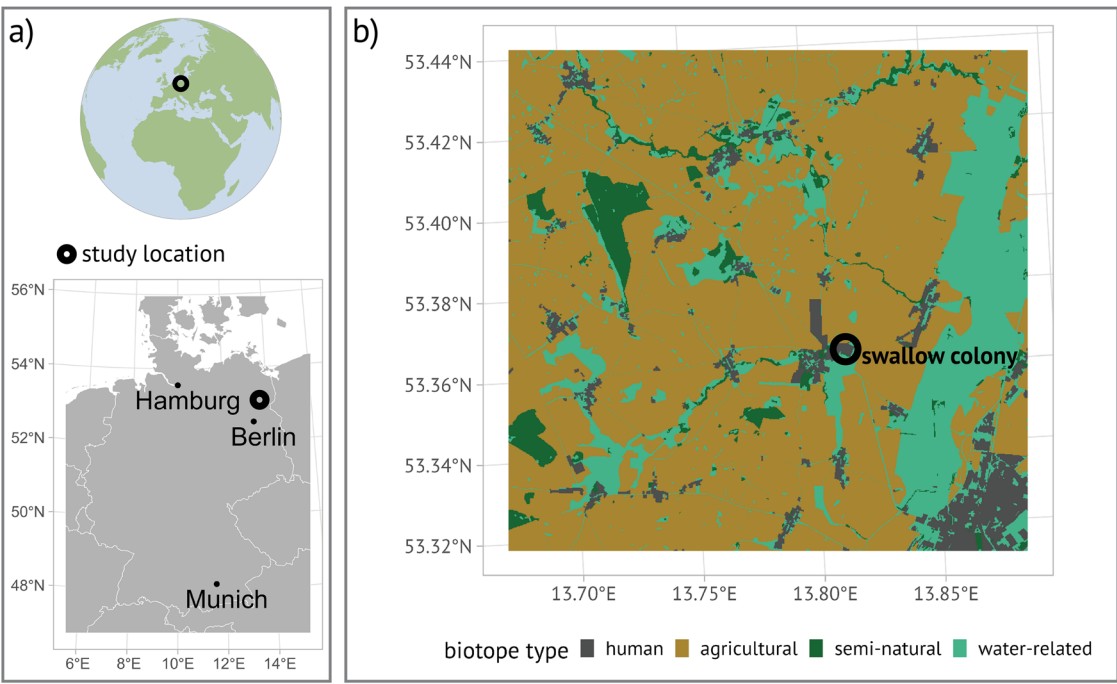

**Fig. 4 | Study area. a** Study location (black circle) in central Europe (top), northeast Germany (bottom). **b** Collated biotope types and location of swallow colony (black circle).

### Study species

We undertook a study spanning four breeding seasons from 2020 to 2023, focusing on two free-ranging populations of barn swallows (BS) and house martins (HM) during their breeding seasons from May to June, when birds nested at the same dairy farm. Both species are obligate insectivorous and hunt aerial insects during foraging trips close to their stationary breeding colonies[80]. Preliminary research in 2019, involving 53 barn swallows and 17 house martins (Supplementary material S3), unveiled a high prevalence (up to 33% in house martins) of haemosporidian parasite infections, specifically by the genera *Plasmodium*, *Leucocytozoon*, and *Haemoproteus*. This coincided with the presence of potential disease vectors (e.g., mosquitoes such as *Culex pipiens* and louse flies of the family *Hippoboscidae*) at their nesting colonies.

### Capture-mark-recapture study

We applied a structured capture-mark-recapture (CMR) design, in which each primary occasion (year) consisted of two secondary sampling occasions exactly 14 days apart (21st and 23rd calendar week). Birds of both species were captured using mist nets (Ecotone 1014/3, mesh 14 × 14 mm). We minimized capturing biases by placing mist nests directly at the gates of the dairy farms (standardized capturing procedure, 7 hours per capturing event) and maintained an equal sampling intensity. Upon capture, we ringed each individual and took morphological measurements, including tarsometatarsus length (mm), body weight (g), wing length (mm), 8th primary feather length (mm), streamer length (mm), and tail fork depth (mm). We estimated fat scores following Kaiser[81], ranging from 1 (not fat) to 8 (high-fat storage), and estimated muscle scores following Bairlain et al.[82] between 0 (muscle depressed) and 3 (full rounded muscle). For each individual, we calculated individual body condition (weight/tarsometatarsus). Due to the weak sexual dimorphism and monochromatism of both species, sexing was performed based on morphological measurements as per Winkler & Jenni[83]. We selected a random sub-sample of 180 ± 67 (mean ± SD) individuals per year and punctured their brachial vein with a hollow needle. We collected small amounts of blood (approx. 30 μl), stored it in stabilization buffer (Qiagen RNAprotect), and later extracted DNA and RNA to assess blood parasite and *Flavivirus* infection, respectively. We used nested polymerase chain reaction (PCR; Supplementary material S4) to

identify the blood parasite genera[84]. Due to the limited number of birds tagged with ATLAS (see 5.3), we pooled blood parasite infections of all genera and treated infection status as binary (i.e. tested individuals are either non-infected or infected).

For *Flavivirus* screening, we ran a pan-flavivirus reverse transcription–polymerase chain reactions (RT-PCR), targeting the non-structural protein 1 gene[85] and *Alphavirus* screening via pan-alphavirus reverse transcription-polymerase chain reactions (RT-PCR), targeting the non-structural protein 1 gene[86]. Despite analysing co-infections with flaviviruses and alphaviruses, we excluded them from subsequent analyses given the insufficient number of infected individuals for a balanced study design (Supplementary material S4).

### Animal tracking study

We tracked birds using a high-throughput reverse-GPS system[21], employing stationary receivers to gather time of arrival data from deployed radio transmitters, thereby calculating real-time locations with notable accuracy[87]. During the CMR, we equipped a total of 71 house martins (HM) and 25 barn swallows (BS) with ATLAS transmitters. Individuals that were tagged and recaptured after 14 days, were not tagged again. Adhering to animal welfare permit guidelines, we tagged only individuals meeting the minimum body weight of 20 g for house martins and 19 g for barn swallows. Considering the low body weights of both species, we attached ATLAS transmitters with skin-friendly glue between the scapulas where feathers are absent, ensuring unrestrained flight ability (Supplementary material S3). We calculated spatial locations via the ATLAS-internal robust-algorithm, with a chosen sampling interval of 0.125 HZ, i.e. one localization every 8 seconds. The mean runtime before tag-loss or tag-failure was 3.95 days [SD: 3.27 days]. We pre-processed all recorded relocations to minimize localization errors in R 4.2.1[88]. Initially, we corrected irregular time stamps to the true sampling interval of 0.125 Hz and calculated the Euclidean distance and time difference from the preceding observation for rough speed estimates. We excluded all observations with high spatial uncertainty or unrealistic speed estimates (Supplementary material S5). In alignment with the species' biology, we only retained observations recorded between nautical dawn and dusk (sun position 12° below horizon; Supplementary material S5). To ensure reliable movement models, we excluded all individuals (n = 21;

18.6% of all tracked birds) with fewer than 500 recorded observations following this filtering procedure. To reduce potential capture-induced behavioural biases, we omitted the initial six hours of the tracking data, which represents the period in which our mist nets were still operative. We used the non-parametric Welch's test to compare movement speeds between non-infected and infected groups[89]. The data set of tracked animals with complete information on infection status and movement behaviour consists of 60 individuals.

## Statistical analyses

We performed all statistical analyses using R 4.2.1[88], and utilized the NIMBLE programming language[90,91] via R 4.2.1[88] for all MCMC methods (5.4.5).

## Individual movement: Foraging ranges

We cleaned movement data and fitted multiple stochastic continuous-time movement models (ctmm), then performed AIC-based model selection and subsequently obtained autocorrelated kernel density estimates (aKDEs) using the *ctmm* package[92]. By visually inspecting autocorrelation structures via variograms, we ensured all individuals exhibited stationary behaviour during the study period, and assessed model convergence. We defined the 95% and 50% aKDE isopleths as the foraging range and core range, respectively, representing the foraging ranges during parental care movements. Lastly, we simulated missing localizations based on the fitted ctmm at short time lags due to receiver-transmitter malfunction, enabling appropriate autocorrelated movement and localization error models. Furthermore, at large time gaps, we performed path reconstruction in case animals were resting, i.e. we filled resting points at locations when animals did not move and therefore no signal could be detected (Supplementary material S5). We used bias-corrected *Bhattacharyya coefficients* to estimate overlap between all individual aKDEs[93]. Finally, we conducted a χ2-Inverse Gaussian hierarchical analysis[33], to evaluate differences in foraging ranges, both between species and between infected and non-infected individuals.

## Individual movement: Behavioural classification

To evaluate behavioural states, we employed hidden Markov models (HMMs) utilizing the *momentuHMM* R package[94]. Our objective was to distinguish among three key behaviours: resting, foraging, and commuting between foraging grounds and the breeding colony. We fitted three-state generalized HMMs to individual ATLAS tracks, using step lengths and turning angles to evaluate discrete behavioural states. We characterized 'resting' by small turning angles and short step lengths, 'foraging' by moderate to large turning angles coupled with medium step lengths, and 'commuting' by small turning angles in conjunction with large step lengths. For the data stream ($z$) probabilities, we applied a Gamma probability distribution to model step length and a Von Mises probability distribution for turning angles, with the covariate 'infection status' using log link functions. We constrained the lower and upper bounds of the natural scale step length parameters, specifically 'resting' was limited to 20 m ± 20 (mean ± SD) within 8 seconds, aligning with the ATLAS localization error when an animal is stationary. In contrast, 'foraging' was constrained to a lower bound of 40 m (mean), indicative of mean movement speeds of 18 km h$^{-1}$. We also imposed constraints on the natural scale concentration parameters (κ) for turning angles: 'resting' and 'foraging' to κ < 5, and 'commuting' to 2 < κ < 10, indicating a peaked distribution for longer 'commuting' distances and wider turning angles for the other states. We modelled the state-transition formula based on environmental covariates such as distances to kettle holes and human settlements in interaction with the blood infection status. The initial state distribution was determined using a regression formula focused on infection status, anticipating behavioural differences between infected and non-infected individuals. To ensure model convergence and numerical stability, we ran each model 50 times with random initial values, selecting the one with the highest maximum likelihood.

Finally, we applied the Viterbi algorithm to each animal's localization, enabling us to evaluate behaviour at each time step. We used the non-parametric Welch's test to compare behavioural differences between non-infected and infected groups[89].

To test, if we would be able to classify behavioural states without using high-resolution tracking data, we resampled our movement data to 30 minutes intervals and re-fitted HMMs (Supplementary material S8).

## Individual movement: Habitat selection

To evaluate habitat selection, we performed an integrated step selection function iSSF;[95] Our primary focus was on the 'foraging' behaviour, leading us to exclude observations classified as 'resting' and 'commuting'. Recognizing the scale dependency of step selection analyses[96], we resampled individual tracks to one-minute intervals, creating five random steps for each observed step. At the end of each movement step, we extracted the spatial covariates (Fig. 4c) to compare the habitat where an animal travelled with habitat that was available at random steps. We utilized generalized linear mixed models (GLMMs) via *glmmTMB*[97], allowing us to evaluate the impact of various habitat types and individual infection status on habitat selection (as fixed effects). Furthermore, we modelled inter-individual preferences in habitat selection as random effects, i.e., a random intercept per individual (both species), and additional random slopes per individual and habitat type (in barn swallows). Random slope models in House martins did not converge. We modelled the habitat selection response, i.e. observed movement steps, using a Poisson distribution and expressed the results as resource selection strength (RSS) relative to the habitat type 'human settlement' — the nesting location for swallows. To mitigate the potential effects of mis-classification in behavioural states, we conducted a secondary analysis using a step selection function that did not differentiate by behavioural state. Instead, this analysis included all observations, providing a comprehensive view of habitat selection irrespective of the classified behaviour (Supplementary material S9).

## Morphologic trait responses to blood parasites

To test if blood parasite infection results in morphological responses of individuals, we performed analysis of variances between both groups, non-infected and infected individuals. To test if blood parasite infection status is related to any other body traits (Fig. 1e), we performed a Principal Component Analysis[98] on all collected morphological traits and grouped individuals by infection status post-hoc (Supplementary material S10). To test if morphological traits would be altered over the timescale of active parasite infection, we accounted for changes in individual body condition within the 14 days of the CMR for infected and non-infected individuals (Supplementary material S10).

## Demographic responses to blood parasites

To assess the impact of blood parasites on survival, we employed multi-event CMR models[99], allowing for uncertainty in infection state observations instead of excluding individuals not tested from subsequent analyses[100]. We utilized Bayesian inference and Markov chain Monte Carlo (MCMC) algorithms to estimate posterior distributions of model parameters, and utilized the NIMBLE programming language[90,91] via R 4.2.1[88]. We describe the ecological process for individuals at first capture as being in one of two ecological states ($z$): alive and infection status positive ($z = BP_{pos}$ with probability $\pi_1$), or alive and infection status negative ($z = BP_{neg}$ with probability $\pi_2 = 1 - \pi_1$). We modelled the latent ecological state as a categorical random variable:

$$Z_{i,\text{first}} \sim Categorical(\boldsymbol{\pi}) \tag{1}$$

After initial capture, we modelled infection dynamics as a first-order Markov process, enabling state transitions between infection states and incorporating survival probabilities (i.e., a dead state $z = D$) between

primary occasions. Consequently, transitions into subsequent ecological states ($z_t$) depended solely on the current state ($z_{t-1}$):

$$\Gamma = \begin{pmatrix} \phi_{BP_{neg}}(1-\psi_{BP_{negpos}}) & \phi_{BP_{neg}}\psi_{BP_{negpos}} & 1-\phi_{BP_{neg}} \\ \phi_{BP_{pos}}\psi_{BP_{posneg}} & \phi_{BP_{pos}}(1-\psi_{BP_{posneg}}) & 1-\phi_{BP_{pos}} \\ 0 & 0 & 1 \end{pmatrix} \begin{matrix} z_{t-1}=BP_{neg} \\ z_{t-1}=BP_{pos} \\ z_{t-1}=D \end{matrix} \quad (2)$$

with column headers $z_t = BP_{neg}$, $z_t = BP_{pos}$, $z_t = D$.

where $\Phi_{BPneg}$ and $\Phi_{BPneg}$ represent the survival probabilities of susceptible and infected individuals, respectively, $\psi_{BPnegpos}$ is the transition probability from negative to positive, and $\psi_{BPposneg}$ is the probability to clear infection between primary occasions, while dead animals remain in the dead state. Here, we could not differentiate between permanent emigration and death, therefore we estimate apparent survival (hereafter: survival).

Additionally, we characterized the observational process through state-dependent detection probabilities (i.e., altered capture probability in case of infection) and the probability of detecting the infection, given that we did not blood sample each individual. We defined seven different observation events: not detected (0), non-infected capture during one secondary sampling occasion (1), non-infected capture during both secondary sampling occasions (2), infected capture during one secondary sampling occasion (3), infected capture during both secondary sampling occasions (4), infection status unknown captured during one secondary sampling occasion (5), and infection status unknown captured during both secondary sampling occasions (6).

Specific observations, $y_t$, are categorical random variables and we describe the state-dependent observation probabilities at primary sampling occasion t as follows:

$$\Omega = \begin{pmatrix} 1-\sum_{y_t=1}^{6} p(y_t) & \beta_{Test}p_{neg}\left(1-p_{neg}\right)^2 & \beta_{Test}p_{neg^2} & 0 & 0 & (1-\beta_{Test})p_{neg}\left(1-p_{neg}\right)2 & (1-\beta_{Test})p_{neg^2} \\ 1-\sum_{y_t=1}^{6} p(y_t) & 0 & 0 & \beta_{Test}p_{pos}\left(1-p_{pos}\right)^2 & \beta_{Test}p_{pos^2} & (1-\beta_{Test})p_{pos}\left(1-p_{pos}\right)2 & (1-\beta_{Test})p_{pos^2} \\ 1 & 0 & 0 & 0 & 0 & 0 & 0 \end{pmatrix} \begin{matrix} z_t=BP_{neg} \\ z_t=BP_{pos} \\ z_t=D \end{matrix} \quad (3)$$

with column headers $y_t=0$, $y_t=1$, $y_t=2$, $y_t=3$, $y_t=4$, $y_t=5$, $y_t=6$.

where $p_{neg}$ and $p_{pos}$ are the capture probabilities of non-infected and infected animals, respectively, while $\beta_{Test}$ denotes the probability of an animal undergoing blood sampling for subsequent parasite screening via PCR and is fixed to the known proportion of individuals tested in each year.

We fitted separate sets of candidate models for barn swallows and house martins, initiating with a constant model that excluded individual or temporal covariates, testing our hypothesis whether blood parasite infection (as a qualitative categorical predictor) would alter survival or capture probabilities. Subsequently, in addition to infection status, we examined time-varying effects on survival probabilities, incorporated fixed effects based on infection prevalence within the colony, and explored the potential of individual heterogeneity in morphology in explaining variation in survival probabilities (Supplementary material S11).

We used uninformative beta (1,1) priors for all parameters. We ran models with 4 chains, starting at widely dispersed initial values, for 20,000 iterations, including a burn-in period of 3,000. We report all parameter estimates as mean and 89% credible intervals[34]. We assessed chain convergence by using Gelman-Rubin statistics[101]; R-hat was < 1.05 in all parameter estimates. We derived posterior distributions for the differences in survival probabilities based on infection status. This was achieved by subtracting the posterior distribution for infected individuals from that of non-infected individuals. Subsequently, we calculated the proportion of posterior samples <0 as evidence in favour of a negative effect of infection on survival (Supplementary material S13). To test if our sample sizes are sufficient to receive unbiased estimates, we ran simulations to validate our models under different scenarios and accounted for error and bias by comparing model estimates and true value (Supplementary material S14).

## Animal research

We have complied with all relevant ethical regulations for animal use. All animal handling procedures adhered to the protocols no. **2347-22-2019**, as approved by the Brandenburg National Office for Occupational Safety, Consumer Protection and Health, and **Beri-017b-21**, as approved by the Brandenburg National Office for the Environment.

## Reporting summary

Further information on research design is available in the Nature Portfolio Reporting Summary linked to this article.

## Data availability

The movement data used for this study is contained in a movebank repository and available on request (Movebank ID: 3053965481). The capture-recapture data is publicly available at https://doi.org/10.5281/zenodo.13934755[102].

## Code availability

All computer code to reproduce the study is publicly available at https://doi.org/10.5281/zenodo.13934755[102].

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

## Acknowledgements
We thank Moritz Wenzler-Meya for preparing environmental data; Jan Pufelski and Yoav Bartan for technical support; Cara Gallagher for language editing; numerous helpers from Leibniz Institute for Zoo- and Wildlife Research and Technical University Berlin for their support during fieldwork; Katja Havenstein, Dagmar Thierer, and Anke Schmidt for their help during laboratory work; the local supporters in the Uckermark area, especially Agrarprodukte Dedelow and family Menke for their long-term support in accessing swallow colonies; and members of the ZALF experimental station in Dedelow, especially Dr. Gernot Verch and Christin Schulz. We also thank the Minerva Foundation and the Minerva Centre for movement ecology for their persistence in supporting and developing ATLAS. We thank Konstantin Kliemke (Bernhard Nocht Institute) for his excellent support in flavivirus analyses. M.G. was supported by the German Research Foundation (DFG) Research Training Group "BioMove" (DFG-GRK 2118/2). W.U., C.S., F.J., V.R., R.T. and S.K.S. are associated with the DFG Research Training Group "BioMove" (DFG-GRK 2118/1 and 2). ATLAS work was supported by the Minerva Centre for Movement Ecology, the Minerva Foundation, the Adelina and Massimo Della Pergola Chair of Life Sciences to R.N., and grant ISF-965/15 to R.N. and S.T. R.L. is funded by the Federal Ministry of Education and Research of Germany (BMBF) under the project NEED (grant number 01KI2022).

## Author contributions
M.G., W.U., C.L., and S.K.S. designed the study; M.G., C.L., W.U., C.S. and S.K.S. collected data in the field; R.T. led lab experiments of blood parasite analyses; J.F. led lab experiments of sex-PCRs; R.L. led lab experiments of flavivirus analyses; M.G., R.S., V.R. and S.K.S. analysed data for the population responses; M.G. and S.K.S. analysed animal movement data. V.R., N.B., F.J. and S.K.S. provided project supervision; R.N., S.T. and F.J. conducted tracking system operation; M.G. lead the writing of the manuscript with support of S.K.S.; all authors edited the manuscript and made valuable scientific contributions throughout the writing process.

## Funding

## Competing interests
The authors declare no competing interests.
