## [Transparent Peer Review file · Communications Biology]

Sick without signs. Subclinical infections reduce local movements, alter habitat selection, and cause demographic shifts

Corresponding Author: Mr Marius Grabow

Version 0:

Reviewer comments:

Reviewer #1

(Remarks to the Author)

This study provides compelling and rare evidence of associations between infection status, movement behavior and foraging choices, and implications for host survival. I think the study is an exciting, novel and important contribution to the literature – the combination of multi-year, fine-scale movement tracking paired with dynamic information on infection status is methodologically novel, and the findings are of broad interest both from a fundamental and applied perspective. The study is also very clearly written, and the statistical and movement analyses performed are appropriate. I have one major point about interpretation of causality in the results, and then just a few minor clarifying comments below.

The study clearly highlights associations between individual movement and infection status, which the authors interpret as effects of infection on movement (and subsequently survival). In theory, however, there are three ways that these associations could arise

- (1) infection alters movement patterns (the authors' hypothesis)
- (2) individual variation in movement behavior predicts exposure to infection (e.g. spending more time closer to breeding colonies predicts exposure to disease vectors)
- (3) Underlying individual variation in condition predicts covariation in infection and condition (e.g. birds with lower body condition are both more susceptible to infection and move less)

I think the paper would strongly benefit from assessing relative support for each of these hypotheses. For example, you could argue that (3) isn't well supported given your finding of no differences in condition between infected and uninfected birds. Are there lines of evidence from your data to distinguish between hypotheses (1) and (2). For example, are there individuals that changed status from uninfected to infected over the course of the study, and, is it possible to quantify a change in movement patterns of these individuals following infection?

Minor comments

Lines 22-23 – Use of 'however', 'yet' and 'but' in short succession is a lot of negatives, and maybe the link to pandemic preparedness isn't immediately clear. Suggest rephrasing second sentence along the lines of 'Monitoring infection-induced movement changes in wildlife would represent a major breakthrough in zoonotic disease surveillance and pandemic preparedness, but field evidence of the effect of...'.

Line 49 – add 'are' after individuals

Line 56 – Avoid starting paragraph with 'however' at start of sentence – either delete, or expand your point 'In spite of the importance of..., incorporating infections is'

Line 68 – should be 'these limitations'

Line 72 – suggest adding commas after 'behaviour' and before 'and investigate'

Line 74 - I think this should be 'philopatric'

Line 80 - I'm not sure it's generally true that avian malaria infections are primarily acquired during wintering (e.g. studies of North American thrushes show they share more blood parasites with breeding range residents than residents at tropical overwintering sites), and in line 408 you suggest a potential association between presence of vectors at nesting colonies. So this needs some qualification (or removing)

Line 87 – avoid repeating 'which' twice in same sentence

Table 1 – can you clarify the definition of overall infection prevalence and the blood parasite lineage specific prevalences, since the % infected by lineage do not sum to the overall infection prevalence in each row.

Line 152 – should be 'relative' (lower case r)

Line 241 – it seems like some infected individuals cleared infection. Did behavioral differences persist after infection was no longer detected in the blood?

Line 526 – I'm curious as to which of your measures of body condition are likely to be altered over the timescale of active parasite infection, and which are measures of inherent condition unlikely to be influenced by current infection status? For example, I could imagine that scaled fat scores could be influenced by current foraging success while some others (tarsus length?) might have been shaped by early life conditions.

Reviewer #2

(Remarks to the Author)

This paper presents four years of data on the demography and high-resolution movement of one population of barn swallows and one population of house martins that nested at the same dairy farm in northeast Germany. About 40% of the populations were infected with haemosporidian parasites, which were tracked via genetic analysis of blood samples. The key feature of the study is the combination of high-resolution movement data and mark-recapture analysis of the populations which combine to give insights into the effects of infection on behaviour and survival. Infection was associated with changes to habitat use such that agricultural areas were no longer avoided and smaller foraging ranges, and reduced survival.

I think that this is, almost, a beautiful paper. The tracking data are fascinating and provide a rare glimpse into the sublethal effects of infection on wild populations. I fully agree with the authors argument that studies on the hidden effects of infection are scarce and this study presents data of such detail that one can easily see the behavioural consequences of infection in an elegant study system. The mark-recapture analysis nicely presents the demographic consequences of infection, and I find the combination of behavioural data of individuals on the level of every 8 seconds with inter-annual survival of the populations to be a complete and satisfying analysis. I therefore think this paper will be a valuable contribution to disease ecology in particular but also to be of broad interest.

The paper is unnecessarily overreaching to make connections to early warning signals and pandemic preparedness. The title and the abstract make reference to these areas, yet there is no development of the ideas in the actual study, what the signals are, or what it is that they presage. There is considerable literature and active interest in the area of early warning signals and epidemics, however, this paper really isn't about any of that. It also doesn't need any of that to be interesting - the combination of the movement analysis, demographic analysis, and epidemiology is enough to stand on its own. The rationale for the study - that work on the hidden costs of infection in wildlife is scarce - is enough on its own for the study to be exciting and of broad interest. I'd suggest revising to remove the connections to EWS and pandemic preparedness and present the work more plainly and descriptively, such as a title like "Effects of haemosporidian parasites on the movement and demography of barn swallows and house martins at a farm in Northeast Germany". Now that may not sound as exciting as the current title, but I'd argue that it is a better description of the work.

One important conclusion is that "infected individuals ... faced reduced survival" (stated in the abstract). However, the posteriors in figure 3b-d and figures in section S13 do not clearly show that survival is lower for infected individuals. Clearly, there is one year when it was not (house martins in 2023), and the other years do have more posterior density towards reduced survival but it is not as clear cut as concluded in the abstract. More care needs to be taken to not overstate the results.

The terms 'hidden infection' and 'subclinical' are used loosely and interchangeably. It would be good to define them and be clear how they differ from conventional definitions or characteristics of disease. Shedding light on these subclinical effects that are relevant for fitness but perhaps not considered in a clinical disease view is an important contribution of this work.

I think that the data from the four years were pooled for the analysis of the tracking data. I can imagine that there may be year-to-year variability in movement due to environmental variation like warm years, wet years, high food abundance, etc. Should that be a consideration of the analysis and how might it affect the interpretation of the results?

Studies such as this suffer the unavoidable flaw that infected and uninfected individuals were not assigned to their infection status randomly, but rather they acquired their infections (or not) naturally. It is therefore always possible that infection is a

consequence of other underlying fitness deficiencies of the the individual hosts, and it is the fitness deficiencies rather than infection that is the cause of the differences in behaviour and survival. This is a problem with most fieldwork studies of infected vs not-infected hosts and is not a fundamental flaw to this study. However, it needs to be acknowledged and discussed, and allowance made for the role of more direct experimental test (e.g. infection challenges) albeit with their own pros and cons.

If I understand correctly, the models used to analyzed movement were fitted separately to each individual, and then nonparametric tests (Welch's) of estimated parameters from those fits were applied to evaluate differences between infected and uninfected groups. This two-step procedure is not ideal as the uncertainty in the estimates from the first step is not passed through to the analysis in the second step. Ideally a model would be fitted to all the data simultaneously and the differences between infected and uninfected groups would be estimated directly from the movement data. This would be a hierarchical movement model, with some parameters for individuals as random effects and fixed effects for the groups. I do not know if such a thing is possible, and it may be beyond the scope of this work. However, it should be considered, and so should the implications of not propagating the uncertainty in the estimates through to the comparison of groups.

The result that infected individuals may frequent agricultural areas is very interesting from the perspective of pathogen transmission between wild and domestic animals. For example there is the current avian flu epidemic. There is also this compelling study of changes in bat behaviour that favours spill-over (Eby, P., Peel, A.J., Hoegh, A. et al. Pathogen spillover driven by rapid changes in bat ecology. *Nature* 613, 340–344 (2023). <https://doi.org/10.1038/s41586-022-05506-2>)

The results for foraging ranges that are stated on lines 153 to 155 are confused. The numbers don't match the results in table 2 or the plots in figure 3A.

I am confused by the results on habitat selection reported in lines 189-201. It needs to be clarified that beta is the estimated coefficient for each habitat type, and define what the value in the +/- refers to (SE, 2SE, 95%CI). Also need to clarify what the p-values that are reported refer to. I think this is a statistical test that the coefficient is different from zero. It would be worth considering putting these results into a table. Another issue is the use of p-values here whereas other analyses use Bayesian methods. It would be nice to be consistent in the statistical approach.

Figure 1d could use a scale bar and the purpose of the boxes labelled "habitat selection water" and "habitat selection agriculture" is not clear.

Section S13 of the supplement - Evidence in favour of lower survival of infected individuals. This section needs an explanation for the calculation used to generate the graphs as well as a more descriptive figure caption. The following caption is repeated in the four figures in the section "Evidence in favour for lower survival of infected individuals in..." but it is not clear what the evidence actually is (i.e. it is a calculation from some of the posteriors of the fitted model).

It is interesting to see the use of the 89% CI . However, many of the results are also presented as 95% CI and 50%CI. It would help the paper to be consistent with the CIs reported.

Figures and tables throughout - please clearly indicate what the points and error bars represent - e.g. mean and SE or median and 95% Credible Interval. For tables please indicate what clearly what the values and CI represent (sometimes this is unstated).

Line 141 what are displacement distances? 24842m seems to be a very long distance for this study.

Line 380 ... can manifest in REDUCED daily foraging ranges...

The last sentence of the conclusion is weak. Delete?

There is some literature on sublethal and indirect effects of infection from studies on salmon that could be useful to broaden the context. A common one is

Miller, K.M., Teffer, A., Tucker, S., Li, S., Schulze, A.D., Trudel, M. et al. (2014). Infectious disease, shifting climates, and opportunistic predators: cumulative factors potentially impacting wild salmon declines. *Evolutionary Applications*, 7, 812-855.

Throughout the paper I would prefer "barn swallows" and "house martins" to BS and HM. While the acronyms save some space it is nicer to read about sparrows than BS!

The writing has many instances where it could be improved by editing of an English language expert.

Reviewer comments:

Reviewer #1

(Remarks to the Author)

The revised manuscript deals comprehensively with all of my questions and concerns, including additional discussion of alternative hypotheses for the observed infection-movement associations and knowledge gaps for future work, as well as undertaking some additional analyses related to changes in condition with infection. Together these changes have further improved an already exciting and well-written manuscript. I have no further comments and think the manuscript is acceptable for publication in its current form.

Reviewer #2

(Remarks to the Author)

I have enjoyed reading the revised manuscript. The authors have done well in considering the comments I made on the first draft and providing thoughtful responses and appropriate revisions to the manuscript.

Response to reviewers:

We would like to thank both reviewers for taking the time to critically assess our work. Their comments were highly insightful, and we believe that they helped us to increase the quality of our manuscript substantially. In the following, we address each of their concerns point-by-point. We have highlighted our revisions in the manuscript in blue italic font.

Reviewers' comments:

Reviewer #1 (Remarks to the Author):

This study provides compelling and rare evidence of associations between infection status, movement behavior and foraging choices, and implications for host survival. I think the study is an exciting, novel and important contribution to the literature – the combination of multi-year, fine-scale movement tracking paired with dynamic information on infection status is methodologically novel, and the findings are of broad interest both from a fundamental and applied perspective. The study is also very clearly written, and the statistical and movement analyses performed are appropriate. I have one major point about interpretation of causality in the results, and then just a few minor clarifying comments below.

R1.1: The study clearly highlights associations between individual movement and infection status, which the authors interpret as effects of infection on movement (and subsequently survival). In theory, however, there are three ways that these associations could arise

- (1) infection alters movement patterns (the authors' hypothesis)**
- (2) individual variation in movement behavior predicts exposure to infection (e.g. spending more time closer to breeding colonies predicts exposure to disease vectors)**
- (3) Underlying individual variation in condition predicts covariation in infection and condition (e.g. birds with lower body condition are both more susceptible to infection and move less)**

I think the paper would strongly benefit from assessing relative support for each of these hypotheses. For example, you could argue that (3) isn't well supported given your finding of no differences in condition between infected and uninfected birds. Are there lines of evidence from your data to distinguish between hypotheses (1) and (2). For example, are there individuals that changed status from uninfected to infected over the course of the study, and, is it possible to quantify a change in movement patterns of these individuals following infection?

Agree – revised. This is an important point, and we fully agree that there are several mechanisms that could explain the observed pattern, given the correlational nature of this study. In the discussion, we have added a paragraph that explores these different hypotheses. While there is evidence supporting our proposed hypothesis, we emphasize that additional experimental evidence is needed to establish greater certainty:

“While our findings provide valuable insights into the relationship between parasitic infections and movement patterns, it is important to consider the potential causal mechanisms underlying these associations. In our correlational study, there are three potential pathways for how these

associations could emerge: (1) infection alters movement patterns, (2) underlying fitness deficiencies cause both reduced survival and movement, and (3) individual movement behaviours predict exposure to infection. We argue that the first causal relationship—infection altering movement patterns—is strongly supported due to the nature of the host-parasite interaction. Avian blood parasites hinder oxygen transport to tissues, resulting in anaemia and physiological limitations, as documented in previous experimental studies. There is limited support for the second hypothesis, which posits that underlying fitness deficiencies predict both reduced survival and movement, as we found no significant differences in morphological traits between infected and non-infected individuals. The third hypothesis, however, that movement behaviours predict exposure to infection, warrants further investigation. One argument against this hypothesis is that individuals typically rest at their colonies, where a dilution effect could potentially reduce individual infection risk, although current evidence of the dilution hypothesis in avian haemosporidians remains inconclusive. On the other hand, swallows, as aerial insectivores, may directly prey on potential disease vectors, potentially lowering the infection risks close to their colonies. Further evidence to discriminate between the first and the third hypothesis could have been collected by comparing the movements of individuals that changed their infection status and were re-tagged. Unfortunately, we did not observe such cases, further highlighting the need for experimental research to test these hypotheses more rigorously. For example, randomly assigning individuals into different treatment groups, performing infection challenges, and assessing movement behaviour. Such experiments could conclusively determine whether movement behaviour is impacted by infection, though they come with ethical considerations and feasibility challenges.” [citations removed for readability] (now line 332-355 in clean version without Marked Up)

Minor comments

R1.2: Lines 22-23 – Use of ‘however’, ‘yet’ and ‘but’ in short succession is a lot of negatives, and maybe the link to pandemic preparedness isn’t immediately clear. Suggest rephrasing second sentence along the lines of ‘Monitoring infection-induced movement changes in wildlife would represent a major breakthrough in zoonotic disease surveillance and pandemic preparedness, but field evidence of the effect of...’.

Agree – revised. We rephrased the first sentences of the introduction and removed the link to pandemic preparedness:

“In wildlife populations, parasites often go unnoticed, as infected animals appear asymptomatic. However, these infections can subtly alter behaviour. Field evidence of how these subclinical infections induce changes in movement behaviour is scarce in free-ranging animals, yet it may be crucial for zoonotic disease surveillance.” (now line 22-25 in clean version without Marked Up)

R1.3: Line 49 – add ‘are’ after individuals

Agree – revised. We have added ‘are’ in the sentence (now line 51 in clean version without Marked Up)

R1.4: Line 56 – Avoid starting paragraph with ‘however’ at start of sentence – either delete, or expand your point ‘In spite of the importance of..., incorporating infections is’

Agree – revised. We have added ‘In spite of the ecological importance of host-parasite dynamics, ...’ (now line 57 in clean version without Marked Up)

R1.5: Line 68 – should be ‘these limitations’

Agree – revised. We have added ‘these limitations’ (now line 70 in clean version without Marked Up)

R1.6: Line 72 – suggest adding commas after ‘behaviour’ and before ‘and investigate’

Agree – revised. We have added both commas (now line 74 in clean version without Marked Up)

R1.7: Line 74 - I think this should be ‘philopatric’

Agree – revised. We have changed ‘philopatry’ to ‘philopatric’ (now line 76 in clean version without Marked Up)

R1.8: Line 80 - I’m not sure it’s generally true that avian malaria infections are primarily acquired during wintering (e.g. studies of North American thrushes show they share more blood parasites with breeding range residents than residents at tropical overwintering sites), and in line 408 you suggest a potential association between presence of vectors at nesting colonies. So this needs some qualification (or removing)

Agree – revised. It is true, studies find contrasting evidence where infection is acquired. Given that we do not have a detailed knowledge on where these individuals were infected, we deleted this sentence to avoid confusing statements.

R1.9: Line 87 – avoid repeating ‘which’ twice in same sentence

Agree – revised. We have removed one ‘which’ in this sentence (now line 87-88 in clean version without Marked Up)

R1.10: Table 1 – can you clarify the definition of overall infection prevalence and the blood parasite lineage specific prevalences, since the % infected by lineage do not sum to the overall infection prevalence in each row.

Agree – revised. Unfortunately, there were multiple mistakes in this table. We have corrected the numbers. Moreover, we now clarify how the overall prevalences are calculated in the caption of Table 1:

*“[...] The overall infection prevalence indicates the percentage of the population infected with any parasite, noting that some individuals had co-infections with the same or different parasite genera, which may cause overall prevalence to differ from the sum of individual genus prevalences (as indicated by *).”*

Accordingly, we included the asterix () in the table whenever the overall infection prevalence is not a sum of the different prevalences, indicating co-infection.*

R1.11: Line 152 – should be ‘relative’ (lower case r)

Agree – revised. Changed to ‘relative’ (now line 153 in clean version without Marked Up)

R1.12: Line 241 – it seems like some infected individuals cleared infection. Did behavioral differences persist after infection was no longer detected in the blood?

Agree – revised. We agree that this is an intriguing question, and addressing it would offer a more nuanced understanding of how disease affects movement, including possible lagged effects on the movement behaviour. However, our study was constrained by the fact that we did not record movement data from the same individuals across both infection statuses. The tracked individuals were a subset of those involved in the CMR study, where we observed changes in infection status among some individuals. However, the chances of tracking enough individuals that underwent a status change were limited, especially due to permit restrictions on tagging individuals below a minimum weight. We put considerable effort into capturing swallows within the CMR study and movement data collection. Unfortunately, due to the observational nature of our field study, we were not able to recapture all individuals and, consequently, could not collect detailed movement data for all of them. To fully address this question, an experimental study would be necessary, which we now discuss (see also R1.1):

“Further evidence to discriminate between the first and the third hypothesis could have been collected by comparing the movements of individuals that changed their infection status and were re-tagged. Unfortunately, we did not observe such cases, further highlighting the need for experimental research to test these hypotheses more rigorously” (now line 348-352 in clean version without Marked Up).

R1.13: Line 526 – I’m curious as to which of your measures of body condition are likely to be altered over the timescale of active parasite infection, and which are measures of inherent condition unlikely to be influenced by current infection status? For example, I could imagine that scaled fat scores could be influenced by current foraging success while some others (tarsus length?) might have been shaped by early life conditions.

Agree – revised. This is a very interesting point and an intriguing analysis that was not considered in the first version of the manuscript. We now performed this analysis to assess alterations of individual body traits within the two weeks of sampling in the same year. We expected that infected individuals would have higher loss of body weight / body condition because their energetic demands should be higher. Likewise, we expected that non-infected individuals should maintain their body weight / body condition within these two weeks, or at least not decrease more significantly than those of infected individuals. However, we found no differences in individuals maintaining their body weight / body condition across the study period regarding their infection status. We have now added this analysis in text and supplementary material (including a new plot in S10):

Methods:

“To test if morphological traits would be altered over the timescale of active parasite infection, we accounted for changes in individual body condition within the 14 days of the CMR for infected and non-infected individuals (Supplementary material S10).” (now line 569-572 in clean version without Marked Up)

Results:

“Furthermore, we found no evidence that morphological traits would be altered over the timescale of active parasitic infection in individuals, as both groups showed only minor alterations in morphological traits over the period of the active parasitic infection (Supplementary material S10).” (now line 220-224 in clean version without Marked Up)

Supplementary S10:

“We calculated the difference in body condition (body weight in g / tarsometatarsus length in mm) for each individual that was recaptured exactly 14 days after the initial capture and excluded all individuals if we lacked parasite infection data. In both species and both infection statuses, individuals increased or decreased their body condition (Figure F14). In non-infected barn swallows body condition decreased on average by $-0.004\text{g mm}^{-1} \pm 0.139$ (mean \pm SD), in infected barn swallows body condition increased by $0.103\text{g mm}^{-1} \pm 0.118$ (mean \pm SD). In contrast, in non-infected house martins body condition increased by $0.037\text{mm}^{-1} \pm 0.148$ (mean \pm SD), and decreased in infected house martins by $-0.023\text{mm}^{-1} \pm 0.121$ (mean \pm SD). Effects were neither statistically significant in barn swallows (Welch t-test: $t(7.5946) = -2.777$, $p = 0.081$), nor in house martins (Welch t-test: $t(60) = 1.776$, $p = 0.081$).” (now line 162-171)

Reviewer #2 (Remarks to the Author):

This paper presents four years of data on the demography and high-resolution movement of one population of barn swallows and one population of house martins that nested at the same dairy farm in northeast Germany. About 40% of the populations were infected with haemosporidian parasites, which were tracked via genetic analysis of blood samples. The key feature of the study is the combination of high-resolution movement data and mark-recapture analysis of the populations which combine to give insights into the effects of infection on behaviour and survival. Infection was associated with changes to habitat use such that agricultural areas were no longer avoided and smaller foraging ranges, and reduced survival.

I think that this is, almost, a beautiful paper. The tracking data are fascinating and provide a rare glimpse into the sublethal effects of infection on wild populations. I fully agree with the authors argument that studies on the hidden effects of infection are scarce and this study presents data of such detail that one can easily see the behavioural consequences of infection in an elegant study system. The mark-recapture analysis nicely presents the demographic consequences of infection, and I find the combination of behavioural data of individuals on the level of every 8 seconds with inter-annual survival of the populations to be a complete and satisfying analysis. I therefore think this paper will be a valuable contribution to disease ecology in particular but also to be of broad interest.

R2.1: The paper is unnecessarily overreaching to make connections to early warning signals and pandemic preparedness. The title and the abstract make reference to these areas, yet there is no development of the ideas in the actual study, what the signals are, or what it is that they presage. There is considerable literature and active interest in the area of early warning signals and

epidemics, however, this paper really isn't about any of that. It also doesn't need any of that to be interesting - the combination of the movement analysis, demographic analysis, and epidemiology is enough to stand on its own. The rationale for the study - that work on the hidden costs of infection in wildlife is scarce - is enough on its own for the study to be exciting and of broad interest. I'd suggest revising to remove the connections to EWS and pandemic preparedness and present the work more plainly and descriptively, such as a title like "Effects of haemosporidian parasites on the movement and demography of barn swallows and house martins at a farm in Northeast Germany". Now that may not sound as exciting as the current title, but I'd argue that it is a better description of the work.

Agree – revised. The previous version of our manuscript had multiple instances where it would oversell the study's contributions, hence, we have revised the introduction and discussion throughout.

Regarding the title, we fully agree on your concern about its clarity and the need of a more precise description of the study's content. While we agree that the current title could be more descriptive and focused, we must also adhere to the journal guidelines, which allow for a maximum of 15 words.

To address this, we propose the title "Sick without signs: subclinical infections reduce local movements, alter habitat selection, and cause demographic shifts". We believe this title captures the core findings of our study while indicating the direction of the found effects. Furthermore, including 'sick without signs' enables this study for researchers that are not familiar with pathogen / parasite related terms.

We believe this title accurately reflects the study's contributions and maintains an engaging narrative without overextending its claims. We hope this revision aligns with your suggestion for a precise and descriptive title.

R2.2: One important conclusion is that "infected individuals ... faced reduced survival" (stated in the abstract). However, the posteriors in figure 3b-d and figures in section S13 do not clearly show that survival is lower for infected individuals. Clearly, there is one year when it was not (house martins in 2023), and the other years do have more posterior density towards reduced survival but it is not as clear cut as concluded in the abstract. More care needs to be taken to not overstate the results.

Agree – revised. To not overstate the results and accounting for the variation among years, we have adjusted the abstract, including a quantitative statement about the average differences in survival probabilities:

"Here, we show that infected individuals had reduced foraging ranges, foraged in lower quality habitats, and faced a lowered survival probability, with an average reduction of 7.4%, albeit with some variation between species and years" (now line 28-31 in clean version without Marked Up)

In the discussion, we have added a statement regarding the uncertainty of the survival estimates, including the specific year when infected house martins indeed had higher survival probabilities compared to non-infected conspecifics:

"Likewise, our analyses of survival rates require caution although our simulation study indicated that our models were capable of obtaining relatively unbiased estimates given the sample sizes. The evidence in favour of lower survival of infected individuals varied between years, with one year even yielding higher survival probabilities of infected individuals (house martins in 2023). Nevertheless,

increased survival may still compromise fitness if infected individuals forego reproduction to enhance immediate survival, potentially diminishing overall fitness. However, since we lack data on reproduction, this hypothesis requires further investigation.” (now line 388-394 *in clean version without Marked Up*)

R2.3: The terms ‘hidden infection’ and ‘subclinical’ are used loosely and interchangeably. It would be good to define them and be clear how they differ from conventional definitions or characteristics of disease. Shedding light on these subclinical effects that are relevant for fitness but perhaps not considered in a clinical disease view is an important contribution of this work.

Agree – revised. To avoid confusion from interchangeably used terms, we have removed all instances of "hidden" infection throughout the manuscript and have used the term "subclinical" infection consistently throughout. We chose to use this term as it is more commonly used in the literature. Additionally, we have expanded the first part of the introduction to include a brief definition:

“As a result, these subclinical infections—where pathogens are present without causing apparent symptoms—can still significantly affect an animal's fitness and may persist for a long time before symptoms become noticeable.” (now line 38-40 *in clean version without Marked Up*)

Please note that the title has also been changed according to R2.1, now referring to "subclinical" instead of “hidden” infection, too.

R2.4: I think that the data from the four years were pooled for the analysis of the tracking data. I can imagine that there may be year-to-year variability in movement due to environmental variation like warm years, wet years, high food abundance, etc. Should that be a consideration of the analysis and how might it affect the interpretation of the results?

Agree – not revised. We fully agree that this would be an interesting and important question to assess the effects of yearly covariates, e.g. climatic variables, and how they could explain specific movement patterns. However, given the small sample sizes of tracked individuals in 2020 and 2021 (see Table 1), such analyses would only make sense for the years 2022 and 2023. We did perform this analysis to the point of autocorrelated kernel density estimators and could confirm that there is indeed some variation in movement behaviour between both years for barn swallows but not for house martins. In all cases, infected individuals moved less. Nevertheless, we chose to not incorporate this analysis in the manuscript acknowledging its weaknesses due to the differences in sample sizes (between years), the differences in prevalence between 2023, and a lack of access to detailed environmental covariates, such as prey abundance.

R2.5: Studies such as this suffer the unavoidable flaw that infected and uninfected individuals were not assigned to their infection status randomly, but rather they acquired their infections (or not) naturally. It is therefore always possible that infection is a consequence of other underlying fitness deficiencies of the the individual hosts, and it is the fitness deficiencies rather than infection that is the cause of the differences in behaviour and survival. This is a problem with most fieldwork studies of infected vs not-infected hosts and is not a fundamental flaw to this study.

However, it needs to be acknowledged and discussed, and allowance made for the role of more direct experimental test (e.g. infection challenges) albeit with their own pros and cons.

Agree – revised. We fully agree that this point – also raised by reviewer 1 (see R1.1) - was not fully developed in the previous manuscript version. Therefore, we included another paragraph into the discussion that elaborates on different scenarios that could lead to the observed pattern, briefly discussing the evidence for each hypothesis.

“While our findings provide valuable insights into the relationship between parasitic infections and movement patterns, it is important to consider the potential causal mechanisms underlying these associations. In our correlational study, there are three potential pathways for how these associations could emerge: (1) infection alters movement patterns, (2) underlying fitness deficiencies cause both reduced survival and movement, and (3) individual movement behaviours predict exposure to infection. We argue that the first causal relationship—infection altering movement patterns—is strongly supported due to the nature of the host-parasite interaction. Avian blood parasites hinder oxygen transport to tissues, resulting in anaemia and physiological limitations, as documented in previous experimental studies. There is limited support for the second hypothesis, which posits that underlying fitness deficiencies predict both reduced survival and movement, as we found no significant differences in morphological traits between infected and non-infected individuals. The third hypothesis, however, that movement behaviours predict exposure to infection, warrants further investigation. One argument against this hypothesis is that individuals typically rest at their colonies, where a dilution effect could potentially reduce individual infection risk, although current evidence of the dilution hypothesis in avian haemosporidians remains inconclusive. On the other hand, swallows, as aerial insectivores, may directly prey on potential disease vectors, potentially lowering the infection risks close to their colonies. Further evidence to discriminate between the first and the third hypothesis could have been collected by comparing the movements of individuals that changed their infection status and were re-tagged. Unfortunately, we did not observe such cases, further highlighting the need for experimental research to test these hypotheses more rigorously. For example, randomly assigning individuals into different treatments groups, performing infection challenges, and assessing movement behaviour. Such experiments could conclusively determine whether movement behaviour is impacted by infection, though they come with ethical considerations and feasibility challenges.” [citations removed for readability] (now line 332-355 in clean version without Marked Up)

R2.6: If I understand correctly, the models used to analyzed movement were fitted separately to each individual, and then nonparametric tests (Welch’s) of estimated parameters from those fits were applied to evaluate differences between infected and uninfected groups. This two-step procedure is not ideal as the uncertainty in the estimates from the first step is not passed through to the analysis in the second step. Ideally a model would be fitted to all the data simultaneously and the differences between infected and uninfected groups would be estimated directly from the movement data. This would be a hierarchical movement model, with some parameters for individuals as random effects and fixed effects for the groups. I do not know if such a thing is possible, and it may be beyond the scope of this work. However, it should be considered, and so should the implications of not propagating the uncertainty in the estimates through to the comparison of groups.

Agree – revised. We agree that a hierarchical model assessing the differences in movement parameters across two groups directly on all individuals would be preferable here, however, we are not aware of the existence of such a method. There were some attempts to incorporate random

effects in Hidden-Markov-models directly but simulation studies found inconclusive results on feasibility and often little advantages at the cost of high computational efforts (McClintock, 2021)*. Random effects in HMMs were further discussed in Glennie et al. (2022)**, who described that such partial pooling – enabling population and individual level parameters – indeed represents a typical pitfall in modelling animal movement, because it can lead to overfitting in specific individuals, leading to wrong conclusions at the population level.

We briefly discuss this in the discussion now: “While our study provides valuable insights into behavioural states and step selection, it is important to acknowledge the limitations of our modelling approach. Specifically, we performed the analyses in two steps, by first fitting HMMs and then applying a Welch test to assess whether the infected and non-infected individuals differ in their behavioural states. Such a two-step procedure cannot fully propagate the uncertainty associated with the parameter estimates obtained from the HMM. Ideally, a hierarchical model should be used to assess behavioural states at the individual level and directly compare them between the infection groups. However, current methods for incorporating random effects into HMMs are computationally demanding and carry the risk of overfitting. Similarly, recent advancements that integrate HMMs with step selection functions (SSFs) into a combined framework can account for uncertainty in behavioural states and state-dependent habitat selection, but these approaches also come with significant computational costs. This highlights the need for continued advancements in developing statistical modelling frameworks to analyse animal movement data.” [citations removed for readability] (now line 404-416 in clean version without Marked Up)

*McClintock, B. T. (2021). Worth the effort? A practical examination of random effects in hidden Markov models for animal telemetry data. *Methods in Ecology and Evolution*, 12(8), 1475–1497.

** Glennie, R., Adam, T., Leos-Barajas, V., Michelot, T., Photopoulou, T., & McClintock, B. T. (2023). Hidden Markov models: Pitfalls and opportunities in ecology. *Methods in Ecology and Evolution*, 14, 43–56.

R2.7: The result that infected individuals may frequent agricultural areas is very interesting from the perspective of pathogen transmission between wild and domestic animals. For example there is the current avian flu epidemic. There is also this compelling study of changes in bat behaviour that favours spill-over (Eby, P., Peel, A.J., Hoegh, A. et al. Pathogen spillover driven by rapid changes in bat ecology. *Nature* 613, 340–344 (2023). <https://doi.org/10.1038/s41586-022-05506-2>)

Agree – revised. This is an important point, which we now briefly introduce in the discussion:

“The finding that infected individuals more frequently utilize agricultural lands is particularly interesting from a One Health perspective. Although avian haemosporidian infections are not pathogenic to livestock, there is growing evidence that spill-over events are becoming more common in human-altered ecosystems, increasing the risk of zoonotic diseases. Notable examples include recent avian influenza outbreaks affecting dairy cows and the zoonotic transmission of Hendra virus from bats to livestock, both eventually affecting humans.” [citations removed for readability] (now line 326-331 in clean version without Marked Up)

R2.8: The results for foraging ranges that are stated on lines 153 to 155 are confused. The numbers don't match the results in table 2 or the plots in figure 3A.

Agree – revised. We introduced errors here while testing different home range size estimators. We have corrected the estimates.

R2.9: I am confused by the results on habitat selection reported in lines 189-201. It needs to be clarified that beta is the estimated coefficient for each habitat type, and define what the value in the +/- refers to (SE, 2SE, 95%CI). Also need to clarify what the p-values that are reported refer to. I think this is a statistical test that the coefficient is different from zero. It would be worth considering putting these results into a table. Another issue is the use of p-values here whereas other analyses use Bayesian methods. It would be nice to be consistent in the statistical approach.

Agree – partially revised. We followed the reviewer's advice and have now presented the results as a table (Table 3). We added a caption that explains the individual columns and the statistical tests employed. While preparing the table, we realized there was a more sophisticated way to construct our model hierarchically, allowing us to better capture inter-individual variation. As a result, we adjusted the model structure. This adjustment does not alter the primary finding—infected individuals more frequently utilize agricultural land—but it does refine some of the other model estimates. More importantly, the new model structure highlights the presence of inter-individual variation in habitat selection, reflecting the complex interaction between host and parasite, as not all individuals are affected uniformly.

These changes have led to updates throughout the manuscript, including the methods, results (including new figure), and discussion.

Regarding the consistency of statistical methods, we fully agree that it would be more coherent to use either Bayesian or frequentist approach, especially because they vary in their philosophy and interpretation. However, the Bayesian framework in the CMR allowed us to construct a hierarchical model that could incorporate individuals with uncertainty in infection status, instead of removing these individuals from our analyses (which would be possible if we stick to the frequentist CMR analyses). Here, a Bayesian framework offers an ideal flexibility for our sampling design. In contrast, all movement models were fitted in frequentist framework, as to our knowledge these methods do not yet exist in Bayesian framework, at least not conveniently embedded in R software.

Given that we incorporated two distinct statistical frameworks in our analysis, we believe it is crucial for readers to easily identify the underlying statistical method within the results section. Therefore, we report all frequentist methods using 95% confidence intervals and corresponding p-values. In contrast, we report all Bayesian methods using 89% credible intervals to clearly indicate that these results stem from a Bayesian approach, which requires a different interpretation compared to frequentist confidence intervals. The choice of the 89% credible interval is deliberate but it aligns with the common default in Bayesian analysis, a practice supported by Kruschke (2014) and McElreath (2018). They advocate for the 89% credible interval for several reasons. Firstly, 95% credible intervals in Bayesian analysis can be less stable, often requiring large effective sample sizes (>10,000) to achieve stability. Secondly, using an 89% credible interval serves as a reminder of the arbitrariness inherent in selecting interval thresholds, and –more importantly - it helps distinguish Bayesian methods from frequentist approaches in a meaningful way.

R2.10: Figure 1d could use a scale bar and the purpose of the boxes labelled “habitat selection water” and “habitat selection agriculture” is not clear.

Agree – revised. We have added a scale bar to Figure 1d, and changed the boxes to “foraging on waterbody” and “foraging on agricultural land”

R2.11: Section S13 of the supplement - Evidence in favour of lower survival of infected individuals. This section needs an explanation for the calculation used to generate the graphs as well as a more descriptive figure caption. The following caption is repeated in the four figures in the section “Evidence in favour for lower survival of infected individuals in...” but it is not clear what the evidence actually is (i.e. it is a calculation from some of the posteriors of the fitted model).

Agree – revised. We now explain in S13 how this evidence is calculated from the posterior distributions:

“To assess the evidence for lower survival in infected individuals, we calculated the difference between all posterior samples from model posterior distributions. Specifically, we combined posterior samples from multiple chains, then computed the difference between the survival probabilities of infected and non-infected individuals. The resulting distribution of these differences allowed us to estimate the proportion of samples with lower survival in infected individuals. We visualized this distribution using density plots, focusing on the proportion of the distribution that falls below zero, which reflects evidence in favour of reduced survival in infected individuals.” (Supplementary material S13)

R2.12: It is interesting to see the use of the 89% CI . However, many of the results are also presented as 95% CI and 50%CI. It would help the paper to be consistent with the CIs reported.

Partially agree – not revised. We report all confidence intervals as 95% CI, all credible intervals as 89% CI (see comment in R2.9). 50% credible intervals were only reported in the CMR analyses to give a more complete overview of the posterior distributions and thus they just accompany the results reported with 89% CIs.

R2.13: Figures and tables throughout - please clearly indicate what the points and error bars represent - e.g. mean and SE or median and 95% Credible Interval. For tables please indicate what clearly what the values and CI represent (sometimes this is unstated).

Agree – revised. We have added the following:

A) in Figure 2: “Throughout panels a-c, points represent mean values, and lines refer to 95% confidence intervals” (now line 213 in clean version without Marked Up)

B) in Figure 3: We have changed the caption of Figure 3:

“Figure 3: Parasite-related demography of two studied host species. Posterior distributions of non-infected and infected barn swallows (BS, left panel) and house martin (HM, right panel); a) Posterior distributions for detection probability. b) Posterior distributions for survival probability. In panels a and b, points represent mean values, and lines refer to 50% (thick line) and 89% credible intervals (thin line). c) Predicted effect of body condition on survival in barn swallows by infection status, according to the selected model. In panel c, thin black lines represent parameter estimates, and shaded areas refer to 89% credible intervals. d) Predicted effect of year on survival in HM by infection status, according to the selected model. In panel d, boxes represent the IQR, lines inside the box the median, and whiskers the 1.5 IQR from the first and third quartiles, respectively. (now line 256-263)

C) *In Table 2: We have added that estimates refer to mean values and estimates brackets refer to 95% confidence intervals*

R2.14: Line 141 what are displacement distances? 24842m seems to be a very long distance for this study.

Agree – revised. We corrected the number and changed the sentence to “mean maximum displacement distances of 2484m ± 1269 (mean ± SD)” (now line 141-142 in clean version without Marked Up)

R2.15: Line 380 ... can manifest in REDUCED daily foraging ranges...

Agree – revised. We have added ‘reduced’ in this sentence (now line 418 in clean version without Marked Up)

R2.16: The last sentence of the conclusion is weak. Delete?

Agree – revised. We have deleted the last sentence of the conclusion.

R2.17: There is some literature on sublethal and indirect effects of infection from studies on salmon that could be useful to broaden the context. A common one is Miller, K.M., Teffer, A., Tucker, S., Li, S., Schulze, A.D., Trudel, M. et al. (2014). Infectious disease, shifting climates, and opportunistic predators: cumulative factors potentially impacting wild salmon declines. *Evolutionary Applications*, 7, 812-855.

Agree – revised. This is indeed an interesting publication. We cite it at multiple instances throughout the manuscript now.

R2.18: Throughout the paper I would prefer “barn swallows” and “house martins” to BS and HM. While the acronyms save some space it is nicer to read about sparrows than BS!

Agree – revised. We have changed all acronyms throughout the text to ‘barn swallows’ and ‘house martins’ respectively. We decided to keep the acronyms in tables and figures where space is limited.

R2.19: The writing has many instances where it could be improved by editing of an English language expert.

Agree – revised. We had a native English-speaking scientist helping us with language editing throughout the manuscript. Her name was added in the acknowledgements, too.